# Dynamic interaction of *MYC* enhancer RNA with YEATS2 protein regulates *MYC* gene transcription in pancreatic cancer

Jayita Roy [ID][1,2], Aniket Kumar [ID][1,2], Shouvik Chakravarty [ID][1,2], Nidhan K Biswas[1,2], Srikanta Goswami [ID][1,2 ✉] & Anup Mazumder [ID][1 ✉]

## Abstract

**Pancreatic ductal adenocarcinoma (PDAC) is one of the most prevalent and aggressive forms of pancreatic cancer with low survival rates and limited treatment options. Aberrant expression of the *MYC* oncogene promotes PDAC progression. Recent reports have established a role for enhancer RNAs (eRNAs), originating from active enhancers, in controlling gene transcription. Here we show that a novel *MYC* eRNA regulates *MYC* gene expression during chronic inflammatory conditions in pancreatic cancer cells. A higher amount of *MYC* eRNA is observed in chronic pancreatitis and in pancreatic cancer patients. We show that *MYC* eRNA interacts with YEATS2, a histone reader protein of the ATAC-HAT complex, and augments the association of YEATS2-containing ATAC complexes with *MYC* promoter/enhancer regions and thus increases *MYC* gene expression. TNF-α induced Tyrosine dephosphorylation of the YEATS domain increases *MYC* eRNA binding to the YEATS2 protein in pancreatic cancer cells. Our study adds another regulatory layer of *MYC* gene expression by enhancer-driven transcription.**

**Keywords** Enhancer RNA (eRNA); *MYC*; YEATS2; Epigenetic Modification; Pancreatic Cancer
**Subject Categories** Cancer; Chromatin, Transcription & Genomics; RNA Biology

## Introduction

Pancreatic cancer is one of the most aggressive cancers with the lowest five-year survival rate (<10%) among common cancers (Orth et al, 2019; Rawla and Barsouk, 2019). It starts with chronic inflammation due to precancerous lesions (Swidnicka-Siergiejko et al, 2017); however, the malignant form, known as pancreatic ductal adenocarcinoma (PDAC) is most prevalent among the patients (>90%) (Adamska et al, 2017; Sarantis et al, 2020). It is

often detected after metastasis or malignancy leading to low survivability of the patient (Lohse and Brothers, 2020). Moreover, the desmoplastic stroma is a hallmark of PDAC giving an advantage of chemotherapy resistance and increasing the dreadfulness of the disease (Pandol et al, 2009; Hosein et al, 2020; Masugi, 2022). So, it is important to elucidate the molecular understanding of PDAC progression for designing novel therapeutic avenues to improve treatment regimen and for better patient survival.

Genomics-based studies have uncovered crucial genetic changes that drive the development of PDAC. These include mutations in oncogenes such as *KRAS* and alterations in tumor-suppressor genes *TP53, CDKN2A, SMAD4, BRCA2*, etc. (Grant et al, 2016; Hayashi et al, 2021). Also, *MYC* is one of such major oncogenes responsible for tumor progression in PDAC (Schleger et al, 2002; Schneider et al, 2021). In pancreatic cancer, c*MYC* lies downstream of the *KRAS* and engages with multiple oncogenic and proliferative pathways (Ala, 2022). Elevated levels of c*MYC* correlate with resistance to chemotherapy, intra-tumor angiogenesis, epithelial-to-mesenchymal transition (EMT), and metastatic progression (Chang et al, 2021). Genomic profiling of PDAC has determined that the *MYC* gene is one of the most commonly amplified genes in PDAC patients (Birnbaum et al, 2011). Nonetheless, it is also reported that gene amplification, transcriptional activation, aberrant cell signaling, or super-enhancer activation are the possible mechanisms for regulating oncogenic *MYC* gene expression (Dong et al, 2020). In cancer cells, *MYC* gene dysregulation is achieved due to the activation of tumor-specific super-enhancers (Fulco et al, 2016). Super-enhancer is a cis-regulatory genomic element comprising of multiple active enhancers where an array of transcription factors may bind (Niederriter et al, 2015; Witte et al, 2015; Wang et al, 2019). Conventionally, enhancers are believed to work by forming a chromatin loop with the promoters (Kadauke and Blobel, 2009). It has been observed that enhancer regions produce noncoding RNAs, which were thought to be transcriptional noise without any regulatory function (Panigrahi and O'Malley, 2021). Later, application of next-generation sequencing (NGS), has shown that these nascent transcripts produced from enhancer regions are tightly regulated (Ghisletti et al, 2010; Kim et al, 2010). So, it is postulated that the enhancer-derived eRNAs might play a significant role in several gene expressions,

[1]Biotechnology Research and Innovation Council-National Institute of Biomedical Genomics (BRIC-NIBMG), Kalyani, West Bengal 741251, India. [2]Regional Centre for Biotechnology, Faridabad, Haryana 121001, India. ✉E-mail: sg1@nibmg.ac.in; am7@nibmg.ac.in

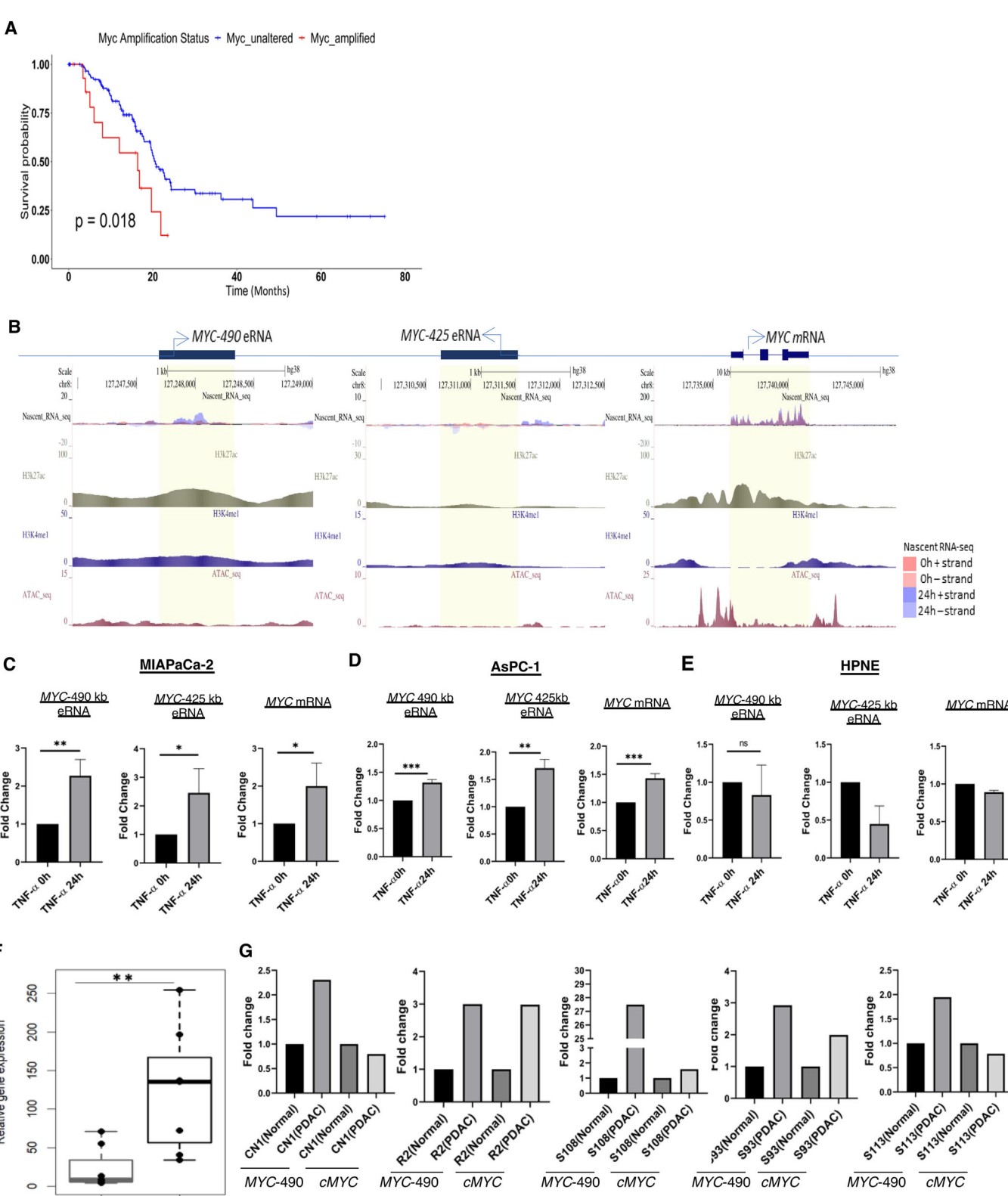

**Figure 1.   Upregulation of *MYC* eRNAs with chronic TNF-α signaling in pancreatic cancer.**

(A) Survival plot analysis depicting the impact of *MYC* amplification status on the overall survival of patients in the TCGA PAAD cohort. (B) UCSC genome browser view showing Nascent RNA-seq signals, ChIP-seq signals for H3K4me1 and H3K27ac and ATAC-seq signals in the genomic regions transcribing *MYC-490*, *MYC-425* eRNA and *MYC* mRNA, respectively in MIAPaCa-2 cells treated with TNFα for 0 h and 24 h. (C) MIAPaCa-2 cells were treated with TNF-α for 0 h and 24 h and RNA expression levels were assessed by RT-PCR. Data are presented as mean ± SD from three independent experiments ($n = 3$). Statistical significance was determined using an unpaired two-tailed $t$ test: \*\*$P = 0.0068$, \*$P = 0.0419$, and \*$P = 0.0483$ for *MYC-490*, *MYC-425* and *MYC* mRNA, respectively. (D) Similarly, RNA from AsPC-1 cells were analyzed by RT-PCR. Results are shown as mean ± SD ($n = 3$); unpaired two-tailed $t$ test: \*\*\*$P = 0.0005$, \*\*$P = 0.0015$ and \*\*$P = 0.0008$. (E) HPNE cells served as the normal control, with results presented as mean ± SD ($n = 3$); unpaired two-tailed $t$ test, ns=not significant. (F) Quantification of *MYC-490-kb* eRNA levels was done in samples obtained from chronic pancreatitis patients ($n = 7$ for normal and $n = 7$ for patients) unpaired two-tailed $t$ test: \*\*$P = 0.0167$. (G) *MYC-490-kb* eRNA as well as *MYC* mRNA levels were measured in pancreatic cancer patients ($n = 5$) and the results were compared with adjacent normal tissue samples. Source data are available online for this figure.

thus regulating important cellular functions (Napoli et al, 2022). However, all enhancers cannot be transcribed. Enhancers with less DNA methylation, proper histone modifications (H3K4me1, H3K27ac), having open chromatin conformation and occupancy by RNA pol II and TFs (TBP, p300/CBP, BRD4, etc.) can synthesize eRNA (Han et al, 2022). eRNAs are relatively less abundant than other cellular RNAs. Though eRNAs are 5'-capped, they are not spliced, and not extensively polyadenylated, thus being vulnerable to exosome-mediated degradation (Hou and Kraus, 2022).

Previous studies have examined the role of *MYC* in the pancreatic cancer super-enhancer network (Dave et al, 2017). However, specific identification of a *MYC* super-enhancer and detailed characterization of *MYC* eRNA function in PDAC have not been comprehensively reported to date. Using the cell-line based model as well as in patient samples, we have found that higher expression of *MYC* eRNA is associated with elevated transcription of the *MYC* gene. Our eRNA overexpression and knockdown experiments have also strengthened these findings. Such observations were absent in pancreatic epithelial cells highlighting cancer-specific activation of super-enhancers. In addition, we have identified a novel interaction of *MYC* eRNA with a histone reader protein YEATS2. The *YEATS2* gene exhibits pronounced amplification across diverse human cancer types (Mi et al, 2017). Given its capacity to recognize modified histones, the YEATS2 protein exerts epigenetic control over the transcriptional program crucial for driving cancer progression. In this study, we have shown that *MYC* eRNA binds with the YEATS2 protein and augments the binding of ATAC-HAT complex to the promoter/enhancer to regulate *MYC* gene transcription. This novel eRNA-protein interaction might be significant in unraveling the functional consequences of *MYC* enhancer-driven RNAs in regulating *MYC* gene expression in PDAC, which can also pave the way for targeting *MYC* gene regulation preventing worse prognosis of pancreatic cancer patients.

## Results and discussion

### *MYC* eRNAs are expressed from super-enhancer regions in pancreatic cancer cells

Activation of the *MYC* gene, that codes for *cMYC* mRNA, is a hallmark of cancer initiation and maintenance (Loven et al, 2013). In various cancer cells, *MYC* dysregulation is achieved through the formation of tumor-specific super-enhancer (Fulco et al, 2016) which is observed for pancreatic cancer too. We have analyzed online available pancreatic adenocarcinoma (PAAD) patient cohort

data ($n = 176$) from the TCGA (The Cancer Genome Atlas) database and found that patients with amplified *MYC* gene have significantly lower survival ($P = 0.018$) than those with unaltered *MYC* gene (Fig. 1A). The lower survival was also found when patient dataset from GSE-62452 database was analyzed (EV1A). In many cases, it has been reported that chronic inflammation in pancreatic cells can lead to pancreatic cancer (Le Cosquer et al, 2023). To mimic the chronic inflammatory condition, we treated MIAPaCa-2, AsPC-1 (which are pancreatic adenocarcinoma cell lines), and HPNE (which is a normal pancreatic epithelial cell line) with TNF-α for 24 h (EV1B–D) as described earlier (Zhao et al, 2016). Then we checked previously published ChIP-seq data for H3K27ac and H3K4me1 from the PDAC cell line as they are key epigenetic features that demarcate active enhancers. We detected high amplitude of the peaks for those two histone marks in *MYC* super-enhancer (SE) regions (Fig. 1B). The ATAC-seq analysis from PDAC cells also correlated with open chromatin regions in those SE upstream of *MYC*. Moreover, we have performed an ex vivo Nascent RNA sequencing from TNF-α treated MIAPaCa-2 cells which showed high levels of enhancer-directed transcription from the *MYC* super-enhancer regions (Fig. 1B). The Nascent RNA-sequencing analysis revealed that *MYC-490-kb* eRNA and *MYC-425-kb* eRNA are part of a group of 19 eRNAs expressed from a 260 kb region (chr8: 127,163,754–127,428,551) of the *MYC* super-enhancer. This region contains clusters of 18 *MYC* enhancers, with lengths ranging from 1 to 20 kb. We selected *MYC-490-kb* and *MYC-425-kb* eRNAs as representative eRNAs from this super-enhancer cluster, as they were specifically upregulated under chronic inflammatory condition. To validate those findings, we further examined those two *MYC* eRNA expression levels in MIAPaCa-2, AsPC-1, and HPNE cells by qPCR analysis. We detected high expressions of *MYC* −490-kb and −425-kb eRNA as well as *MYC* mRNA in both the cancer cell lines (Fig. 1C,D) and HCT-116—a colon cancer cell (EV1E), but not in HPNE (Fig. 1E). Next, we checked *MYC-490-kb* eRNA levels in normal and chronic pancreatitis (CP) patients and detected higher *MYC-490* eRNA in CP patients (Fig. 1F). Moreover, we examined the *MYC-490* eRNA as well as *MYC* mRNA levels in Pancreatic cancer patients and compared them to adjacent normal tissue where *MYC-490-kb* eRNA levels were significantly upregulated in pancreatic cancer tissues (Fig. 1G), thus validating our initial cell-line-based observations.

### *MYC-490-kb* eRNA regulates *MYC* gene transcription

Previous reports have described that the half-life of eRNAs is short, and their functions are mostly confined to the nucleus (Lewis et al, 2019). So,

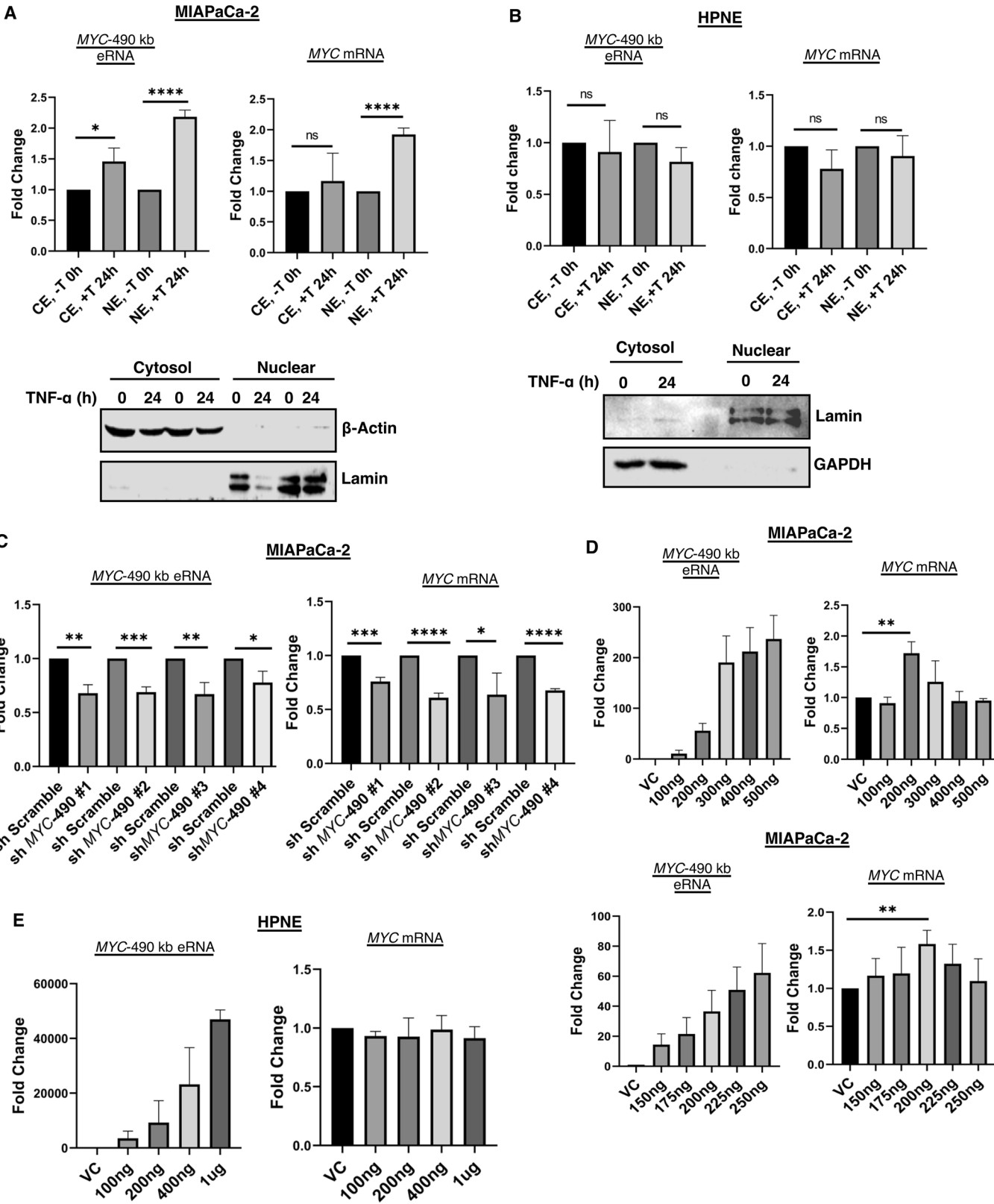

**Figure 2. MYC eRNA plays a regulatory role in modulating MYC gene expression.**

(A) Nascent RNAs were captured from the nuclear and cytosolic fraction using Click-iT kit and analyzed by RT-PCR. Data are presented as mean ± SD from three independent experiments ($n = 3$). Statistical significance was determined using an unpaired two-tailed $t$ test: *$P = 0.0227$, ***$P = 0.0001$ for cytosolic and nuclear fraction, respectively, Nascent MYC mRNAs were also examined under the same conditions. Data are presented as mean ± SD from three independent experiments ($n = 3$). Unpaired two-tailed $t$ test: ns and ****$P = 0.0001$. Western blot analysis for β-Actin and Lamin was done to assess the fractionation of MIAPaCa-2 cells. (B) Nascent MYC eRNA and mRNA in the nuclear and cytosolic fractions were measured in the HPNE cells. Western blot analysis for β-Actin and Lamin was done to assess the fractionation of HPNE cells. Unpaired two-tailed $t$ test: ns. (C) Knockdown of MYC-490-kb eRNA was performed using four different shRNAs and levels of MYC eRNA and MYC mRNA were measured in MIAPaCa-2 cells by RT-PCR. Data are presented as mean ± SD from three independent experiments ($n = 3$). Unpaired two-tailed $t$ test: **$P = 0.0022$, ***$P = 0.0004$, **$P = 0.0059$, *$P = 0.0212$ for MYC-490-kb eRNA and ***$P = 0.0004$, ****$P = 0.0001$, *$P = 0.0348$, ***$P = 0.0001$ for MYC mRNA. (D) MYC-490-kb eRNA were overexpressed in MIAPaCa-2 cells for 48 h and MYC mRNA expression were checked by RT-PCR. Data are presented as mean ± SD from three independent experiments ($n = 3$). Unpaired two-tailed $t$ test: **$P = 0.0025$ for MYC mRNA upper panel and **$P = 0.0051$ for MYC mRNA lower panel. (E) A similar experiment as in (D) was performed in the HPNE cell line to assess any change in MYC mRNA expression. Source data are available online for this figure.

we checked nascent eRNA levels from nuclear fraction of TNF-α treated MIAPaCa-2 cells using Click-chemistry and detected higher MYC-490-kb eRNA levels in nuclear fraction with TNF stimulation (Fig. 2A). We detected very little leaching of nuclear fractions as evaluated by MYC intron levels from both the fractions (EV2A). However, we could not detect any upregulated MYC-490-kb eRNA in nuclear fraction of HPNE cells with same TNF-α stimulation (Fig. 2B). To check whether MYC eRNAs are regulating MYC gene transcription, we performed shRNA-mediated knockdown of MYC-490-kb eRNA (Fig. 2C, left panel) and detected a reduction in MYC gene expression in MYC eRNA KD cells nearly by 30% (Fig. 2C, right panel). To further confirm the fact, we overexpressed MYC-490-kb eRNA in MIAPaCa-2 cells and checked MYC mRNA levels without any TNF-α treatment. At a particular stoichiometric ratio of MYC-490-kb eRNA, MYC mRNA was significantly upregulated in MIAPaCa-2 cells (Fig. 2D, upper panel) but not at all in HPNE cells (Fig. 2E). We performed additional fine-tuning using intermediate plasmid DNA concentrations of MYC-490 eRNA to assess the consistency of MYC mRNA upregulation. Our results again confirmed that MYC mRNA expression peaked at 200 ng DNA overexpression set, whereas intermediate concentrations exhibited a gradual increase in expression levels, though not to the same extent as observed at 200 ng (Fig. 2D, lower panel). These findings indicated that a critical physiological threshold of MYC-490 eRNA is required to upregulate MYC mRNA in PDAC cells. We overexpressed the MYC-425-kb eRNA also in the same stoichiometric ratio but could not observe any MYC gene upregulation (EV2B). These data establish the fact that MYC-490-kb eRNAs were aberrantly expressed and they were regulating MYC oncogene expression in pancreatic cancer cells with chronic inflammation. Previously, it has been shown that two MYC eRNAs transcribed from ESE in lymphoblastoid cells (LCL) regulate MYC gene expression and thus regulate LCL growth and survival (Liang et al, 2016). Our findings are consistent with the recent report that MYC eRNAs have an immense effect in MYC gene transcription. It would be interesting to check how do these MYC eRNAs regulate MYC gene transcription—whether they act individually or multiple MYC eRNAs act together.

## MYC-490-kb eRNA interacts with YEATS2 protein

The discovery of enhancer transcription has uncovered the exciting area of exploring eRNA functions. In the previous section, we have shown that MYC-490-kb eRNAs were regulating MYC oncogene expression but the mechanism was not completely understood. Previous reports have shown that BRD4 or CBP/P300, which primarily function as a histone reader protein, have the capability to engage with various enhancer RNAs (Bose et al, 2017; Rahnamoun et al, 2018). In order to identify the MYC-490-kb

eRNA-interacting proteins, we conducted an in-silico analysis using RPIseq software and HDOCK web server to assess the binding propensity of different histone modifiers with the MYC-490 kb eRNA. In our analysis, we have identified a novel interaction between YEATS2 and the MYC-490 kb eRNA along with some other known eRNA-interacting histone modifiers (EV3A,B). YEATS domain-containing 2 (YEATS2) is a scaffolding subunit of the Ada-two-A-containing (ATAC) complex, which is a well-known conserved metazoan HAT complex. Mi et al have shown that YEATS2 can act as a histone H3K27ac reader and can epigenetically regulate gene expressions required for NSCLC tumorigenesis (Mi et al, 2017). Previously, it was shown that YEATS2 can interact with RNA (He et al, 2016). But what type of RNA can interact with YEATS2 and consequence of YEATS2-RNA interaction was not known.

To validate our in-silico analysis, we checked the MYC eRNA association with YEATS2 by UV-RIP. We detected MYC-490-kb eRNA was interacting more with YEATS2 in TNF-dependent manner in MIAPaCa-2 cells (Fig. 3A) as well as in AsPC-1 cells (Fig. 3B). However, we could not detect any such increase in association of MYC-490-kb eRNA with YEATS2 in HPNE cells (Fig. 3C), although the YEATS2 protein levels were similar in MIAPaCa-2 and HPNE cells (Fig. 3D). Thus, these observations pointed out that some cancer-specific events might be regulating the dynamic interaction of MYC-490-kb eRNA with YEATS2 in PDAC cells. Then we performed YEATS2 knockdown in MIAPaCa-2 cells using four different shRNAs and found a significant reduction in MYC mRNA levels following YEATS2 depletion, highlighting the essential role of YEATS2 in MYC transcription (Fig. 3E). This additional evidence strengthens our conclusion that YEATS2 is a critical regulator of MYC gene transcription and further supports the central claim of our study. Moreover, we checked the association of MYC-490-kb eRNA with BRD4 protein but could not detect any significant increase in association in TNF-stimulated condition (EV3C). We have checked the interaction between YEATS2 and the MYC-425-kb eRNA; however, no significant increase in association was detected (EV3D). These data proved a specific interaction of MYC-490-kb eRNA to YEATS2 protein in pancreatic cancer cells.

YEATS2 has 21 Tyr amino acid residues in the whole protein (EV3E,F), out of which 4 Tyr amino acids are within the YEATS domain and one of the Tyr (Y313) residues is in the 1st RNA binding region (EV3E,F). It has already been reported that Tyr phosphorylation of Argonaut2 protein could interfere with miRNA binding during pro-inflammatory immune response (Mazumder et al, 2013). So, we checked the overall Tyr phosphorylation level in YEATS2 protein with TNF stimulation in MIAPaCa-2, AsPC-1 and HPNE cells. Interestingly, we observed a decrease in Tyr phosphorylation of YEATS2 protein with TNF stimulation in MIAPaCa-2 and AsPC-1 cells (Fig. 3F,G). In the absence of Tyr

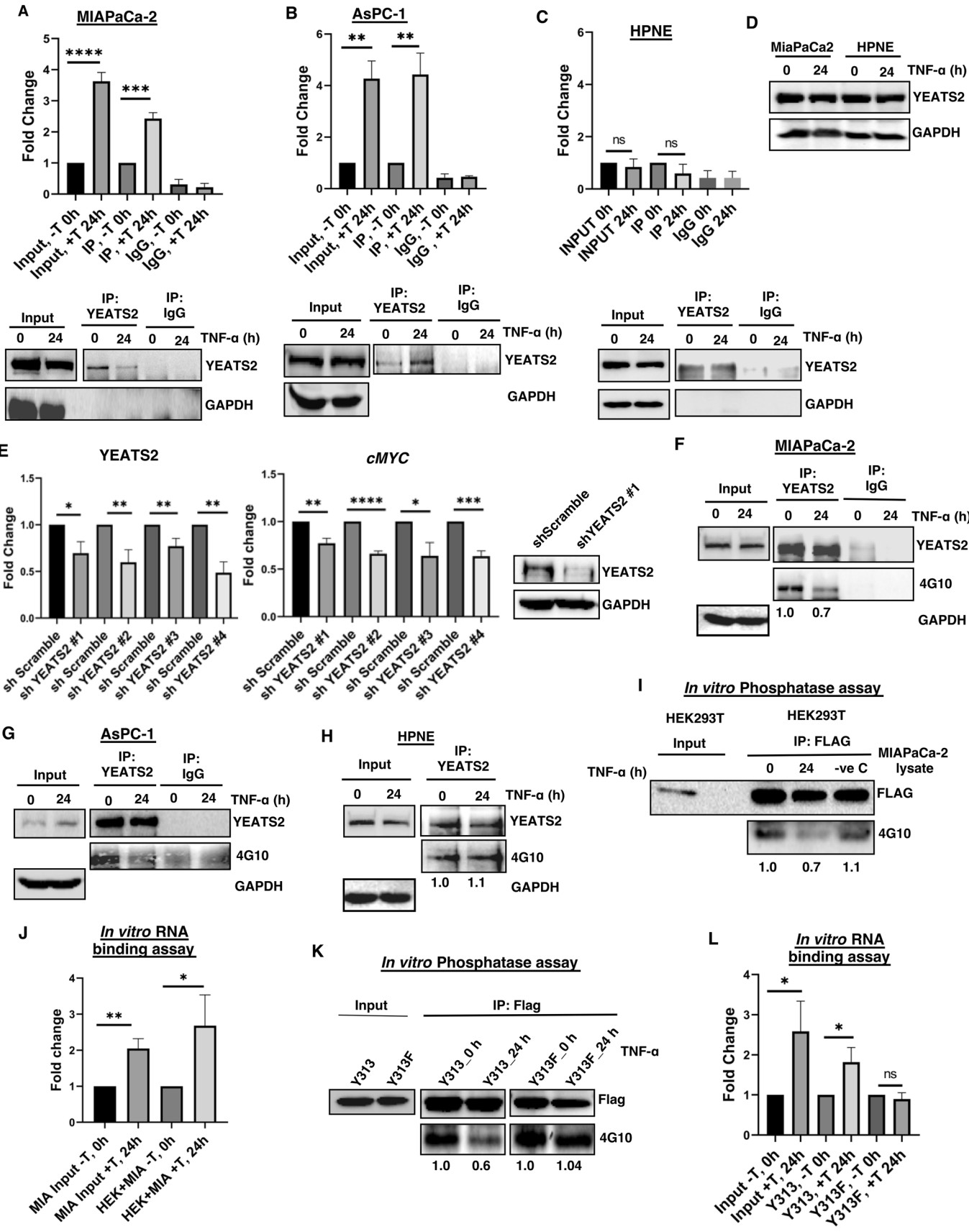

◄

**Figure 3.   Novel interaction of *MYC*-490 eRNA with YEATS2 protein.**

UV-RIP experiment was performed in TNF-α stimulated condition and YEATS2 associated *MYC*-490 eRNA was checked by RT-PCR in (**A**) MIAPaCa-2, data are presented as mean ± SD from three independent experiments ($n = 3$). Statistical significance was determined using an unpaired two-tailed $t$ test: ****$P = 0.0001$ for Input, ***$P = 0.0002$ for IP and (**B**) AsPC-1 cells, data are presented as mean ± SD from three independent experiments ($n = 3$). Unpaired two-tailed $t$ test: **$P = 0.0012$ for Input and **$P = 0.002$ for IP. Western blot was performed to check the successful pulldown of YEATS2 protein. GAPDH served as loading control. (**C**) A similar UV-RIP experiment with YEATS2 antibody was performed for HPNE cells. (**D**) Immunoblot analysis showing YEATS2 protein level in MIAPaCa-2 and HPNE cells. GAPDH served as loading control. (**E**) YEATS2 protein was knocked down by four different shYEATS2 molecules and *cMYC* expression was checked by RT-PCR. Data are presented as mean ± SD from three independent experiments ($n = 3$). Unpaired two-tailed $t$ test: *$P = 0.0129$, **$P = 0.0068$, **$P = 0.009$, **$P = 0.0016$ for YEATS2 and **$P = 0.0017$, ****$P = < 0.0001$, *$P = 0.0109$, ***$P = 0.0004$ for *cMYC*. YEATS2 knockdown was further checked by WB (right panel). GAPDH served as a loading control. (**F–H**) The phospho-tyrosine level of YEATS2 protein was checked by immunoprecipitating YEATS2 from TNF-α stimulated MIAPaCa-2 (**F**), AsPC-1 (**G**) and HPNE cells (**H**) followed by western blotting for 4G10 and YEATS2 antibody. GAPDH served as a loading control. (**I**) In vitro phosphatase assay was performed where YEATS domain was pulled down from HEK293T cells and incubated with MIAPaCa-2 cell lysate treated with TNF-α for 0 and 24 h. Phosphorylation levels were assessed via Western blot analysis using 4G10 antibody. (**J**) A similar approach was taken for an in vitro *MYC* eRNA binding assay to investigate the correlation between tyrosine phosphorylation and *MYC* eRNA binding affinity for the YEATS domain. Data are presented as mean ± SD from three independent experiments ($n = 3$). Unpaired two-tailed $t$ test: **$P = 0.0026$ for Input, *$P = 0.027$ for IP (**K**) An in vitro phosphatase assay was performed, followed by Western blot analysis to evaluate the phosphorylation level of the Y313 and Y313F mutant of YD domain. (**L**) Same experiment as (**J**) was done to investigate the correlation between *MYC* eRNA binding with Y313 or Y313F mutant. Data are presented as mean ± SD from three independent experiments ($n = 3$). Unpaired two-tailed $t$ test: *$P = 0.0219$ for Input, *$P = 0.0177$ for IP. Source data are available online for this figure.

phosphorylation, the propensity of RNA binding to YEATS2 protein will be more due to less charge-charge repulsion between the phosphate group of Tyr and the 5'-Phosphate group of RNA, correlating our previous observation of UV-RIP experiment (Fig. 3A) that TNF induced more *MYC* eRNA association with YEATS2. However, this dynamic change in Tyr phosphorylation was not observed for HPNE cells (Fig. 3H), again pointing out a cancer cell-specific post-translational modification for YEATS2 that regulates *MYC* eRNA binding. Next, we performed in vitro phosphatase assay by overexpressing Flag-tagged YEATS domain (YD) in HEK293T cells, pulled down by Flag bead and incubated the YD-bound bead with TNF-treated or untreated MIAPaCa-2 cell lysate and measured the phospho-tyrosine level by western blotting. This data showed a decrease in phospho-tyrosine level in the Flag-YD (Fig. 3I). Then, we did in vitro RNA binding assay for the Flag-tagged YD and observed higher binding of *MYC*-490-kb eRNA when incubated with TNF-treated MIAPaCa-2 cell lysate (Fig. 3J). To further validate our claim, site-directed mutagenesis was performed to investigate the functional impact of specific amino acid substitutions at position 313 in the YEATS domain. By introducing the Y313F mutations, we aimed to mimic a non-phosphorylated state to elucidate the role of tyrosine 313 phosphorylation in the domain's activity and its overall functional significance. In the in vitro phosphatase assay, no changes in phosphorylation levels were observed for the Y313F mutant (Fig. 3K). Similarly, in the in vitro RNA binding assay, we observed higher binding of the enhancer RNA to the wild-type protein compared to that of mutant protein with TNF stimulation (Fig. 3L). These results suggest that tyrosine 313 in the YEATS domain is pivotal in regulating the phosphorylation-dephosphorylation cycle and is critical for the subsequent binding to *MYC* enhancer RNA, which again confirms our claims.

## *MYC*-490-kb eRNA augments the binding of YEATS2-containing ATAC complex to specific enhancer and promoter region

Since we have seen higher *MYC* eRNA as well as *MYC* gene expression in chronic inflammatory conditions, we checked the association of YEATS2-containing ATAC complex with *MYC* enhancer and promoter region by ChIP-qPCR (Fig. 4A,B). As the

ChIP-grade YEATS2 antibody was not commercially available, we have used ChIP-grade ZZZ3 antibody to examine the ATAC complex occupancy as described earlier (Mi et al, 2017). We observed a significantly higher association of ATAC complex in *MYC*-490-kb (Fig. 4B) and *MYC*-425-kb (EV4A) enhancer region as well as in the *MYC* promoter region with TNF-α stimulation in MIAPaCa-2 cells (Fig. 4B) and in AsPC-1 cells (Fig. 4C) but not in HPNE cells (EV4B). It has been reported that YEATS2 has a higher affinity for crotonylated histones compared to acetylated histones (Li et al, 2016). So, we immediately checked the overall histone crotonylation signal in *MYC* enhancer as well as promoter regions. We have not detected any significant upregulation in histone crotonylation with TNF stimulation (Fig. 4D). We further checked the specific H3K27cr signal in *MYC* enhancer or promoter region (EV4C) but observed no significant change in H3K27cr signal with TNF stimulation. It is well-reported that MYC, as a transcription factor (TF), can regulate its own synthesis by a feed-forward loop (Dang 2013). So, we checked the association of MYC protein in the *MYC* enhancer and promoter regions (EV4D). However, we could not detect any increase in MYC TF occupancy with TNF stimulation. Next, we knocked down *MYC*-490-kb eRNA by shRNA and checked YEATS2-containing ATAC complex occupancy to chromatin by ChIP-qPCR analysis (Fig. 4E). We observed lesser association of ATAC complex with decrease in *MYC*-490 eRNA level in MIAPaCa-2 cells. Also, we did the ChIP-IP for YEATS2 from TNF-stimulated MIAPaCa-2 cells and observed higher H3 association (Fig. 4F, upper panel) with TNF stimulation. Similarly, in the reverse ChIP-IP with histone H3, we detected higher binding of YEATS2 with H3 in TNF-stimulated condition (Fig. 4F, middle panel). However, this association of YEATS2 with Histone H3 was not observed in HPNE cells, again explaining the mechanism behind tissue specificity (Fig. 4F, lower panel). To further confirm the mechanism, we used the similar *MYC*-490-kb eRNA overexpressed system to check the association of YEATS2 with Histone H3 by ChIP-IP. We did the IP with YEATS2 and checked H3 association (Fig. 4G, upper panel) and vice versa (Fig. 4G, lower panel). In both cases, *MYC* eRNA augmented the binding of YEATS2 to Histone H3, thus proving our claim that it was *MYC*-490-kb eRNA that bound to YEATS2 protein and recruited the ATAC complex to *MYC* promoter/enhancer to increase *MYC* gene synthesis in chronic inflammatory conditions.

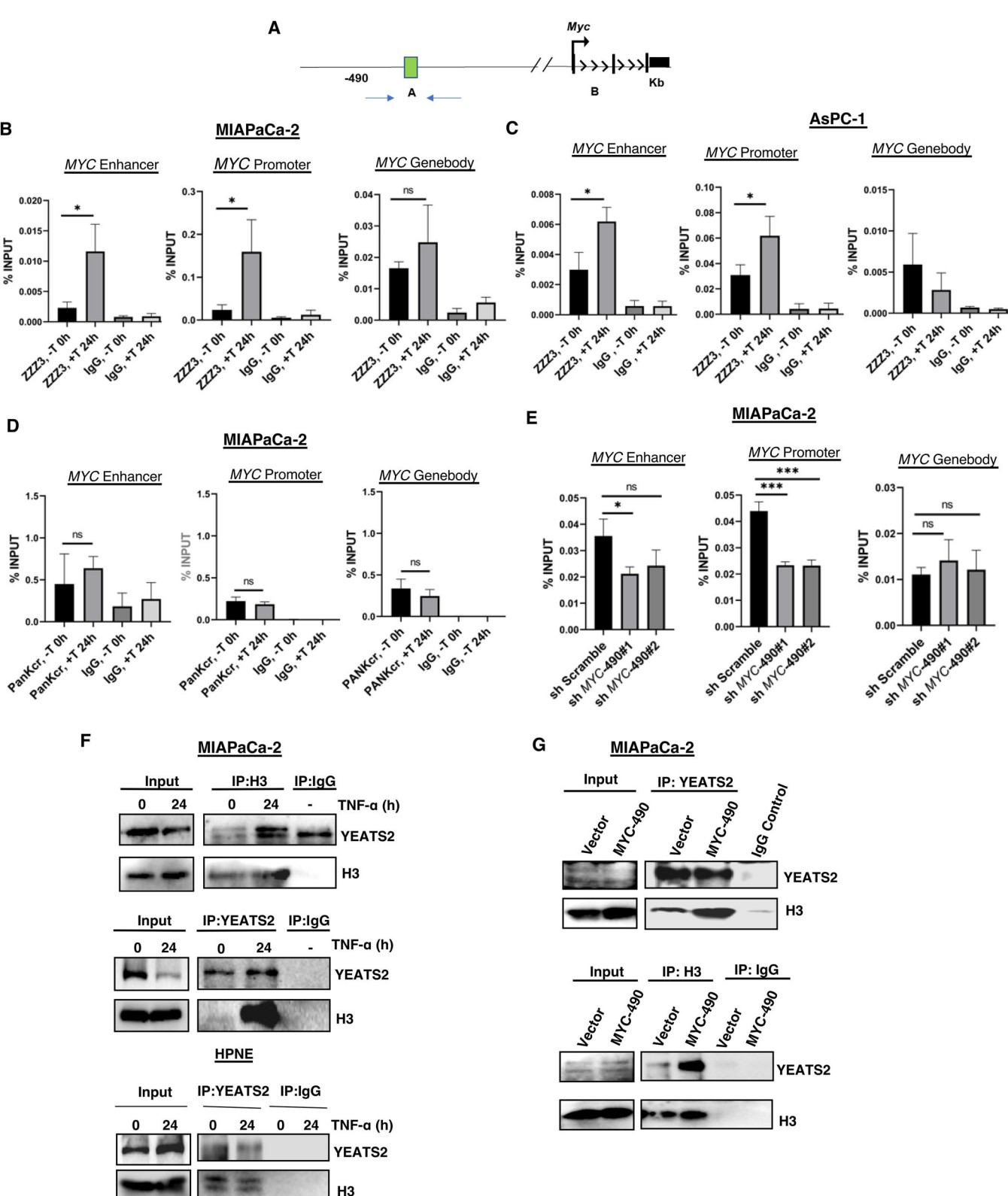

**Figure 4.  Binding of *MYC* eRNA to YEATS2 augments association of YEATS2-containing ATAC complex to *MYC* gene regulatory regions in PDAC.**

(A) Schematic of genomic position of *MYC* gene TSS and its upstream enhancer (490 kb). (B) ChIP-qPCR analysis was performed using ZZZ3 antibody to check YEATS2-containing ATAC complex occupancy in 24 h TNF-α stimulated MIAPaCa-2 cells. Data are presented as mean ± SD from three independent experiments ($n = 3$). Statistical significance was determined using an unpaired two-tailed $t$ test: *$P = 0.0246$, *$P = 0.0363$, ns for *MYC* enhancer, promoter and gene body, respectively. (C) Similar ChIP-qPCR was done from AsPC-1 cells. Data are presented as mean ± SD from three independent experiments ($n = 3$). Unpaired two-tailed $t$ test: *$P = 0.0199$, *$P = 0.036$, ns for *MYC* enhancer, promoter and gene body, respectively. (D) ChIP-qPCR analysis with PanKcr antibody was done from TNF-α stimulated MIAPaCa-2 cells with 0 h and 24 h of activation. Unpaired two-tailed $t$ test: ns. (E) ChIP-qPCR analysis was performed in *MYC*-490 knockdown cell for *MYC* enhancer and promoter regions. Data are presented as mean ± SD from three independent experiments ($n = 3$). Unpaired two-tailed $t$ test: ns and *$P = 0.0231$ for *MYC* enhancer, ***$P = 0.0006$ and ***$P = 0.0009$ for promoter and ns for gene body, respectively. (F) IP was done with Histone H3 antibody from untreated and TNF-α treated MIAPaCa-2 cells and YEATS2 association was checked by WB (upper panel). IP was done with YEATS2 antibody in similar condition in MIAPaCa-2 cells and Histone H3 association was checked by WB (middle panel). IP was done with YEATS2 antibody from TNF-α stimulated HPNE cells and Histone H3 association was checked by WB (lower panel). (G) In *MYC*-490 eRNA overexpressed MIAPaCa-2 cells, IP was done with YEATS2 antibody and its interaction with H3 was checked by western blotting (upper panel). Similar IP was done with Histone H3 antibody and its interaction with YEATS2 protein was checked by WB (lower panel) ($n = 3$). Source data are available online for this figure.

It has been proposed that noncoding RNAs, including enhancer RNAs (eRNAs), may modulate chromatin-associated protein function, potentially influencing the recruitment or activity of factors like YEATS2. However, direct evidence supporting this regulatory mechanism remains limited. Here, we have shown the *MYC*-490-kb eRNAs augment the binding of YEATS2 to histones H3. However, the specific H3 modification that facilitates this interaction remains unknown in our study. Identifying the precise modification could provide critical insights into the mechanisms underlying this binding and its role in enhanced gene regulation.

## *MYC*-490-kb eRNA-mediated *MYC* gene upregulation drives cancer cell progression

MYC is a well-known transcriptional factor that plays various roles in different cancers. The *MYC* eRNA-induced *MYC* gene expression could be another layer of MYC protein-mediated cancer progression. To check that, we performed the cell proliferation by MTT assay in *MYC*-490 overexpressed condition as well as in TNF-stimulated condition (Fig. 5A). We observed an increase in cell proliferation in both the conditions where *MYC* genes were upregulated via *MYC*-490-kb eRNA. We further investigated the cell migration via wound healing assay in similar condition (Fig. 5B). In both situations, we have observed an increased rate of wound healing due to the presence of higher *MYC*-490-kb eRNA in MIAPaCa-2 cells. Taken together, these data correlate the fact that *MYC* eRNA-mediated *MYC* gene amplification in chronic

inflammatory condition might lead to pancreatic cancer progression (Fig. 5C).

Future research should aim to elucidate the exact histone modification responsible, as this knowledge could deepen our understanding of eRNA-mediated chromatin remodeling and transcriptional activation. It would be interesting to check whether YEATS2 binds to other eRNA molecules to regulate a specific set of gene expressions in different conditions in future. Thus, our novel findings could open up a new avenue where YEATS2-eRNA interaction site can be used as a potential therapeutic target to prevent cancer progression in the future.

## Methods

### Cell culture and reagents

Human pancreatic ductal adenocarcinoma (PDAC) MIAPaCa-2 cells and human pancreatic adenocarcinoma AsPC-1 cells were kind gift from Dr. Shantibhushan Senapati, human pancreatic epithelial nestin-expressing (HPNE) cells, and human colon cancer cells, HCT-116 were purchased from ATCC. MIAPaCa-2, AsPC-1 and HCT-116 cells were cultured in DMEM (high glucose) from Gibco, which was supplemented with 10% fetal bovine serum (FBS, Gibco) and 1% penicillin–streptomycin (Pen/Strep, GIBCO).

**Reagents and tools table**

| Reagent/resource | Reference or source | Identifier or catalog number |
|---|---|---|
| **Experimental models** | | |
| MIAPaCa-2 | Kind gift from Dr. Shantibhushan Senapati, BRIC Institute of Life Sciences, Bhubaneswar. | N/A |
| AsPC-1 | Kind gift from Dr. Shantibhushan Senapati, BRIC Institute of Life Sciences, Bhubaneswar. | N/A |
| HPNE | ATCC | CRL-4023 |
| HCT-116 | ATCC | CCL-247 |
| **Recombinant DNA** | | |
| pcDNA3.1 | Addgene | |

| Reagent/resource | Reference or source | Identifier or catalog number |
|---|---|---|
| pIRES-neo | Addgene | |
| pLKO.1-TRC | Addgene | |
| sh*MYC*-490 eRNA #1 | This Study | N/A |
| sh*MYC*-490 eRNA #2 | This Study | N/A |
| sh*MYC*-490 eRNA#3 | This Study | N/A |
| sh*MYC*-490 eRNA #4 | This Study | N/A |
| shYEATS2 #1 | This Study | N/A |
| shYEATS2 #2 | This Study | N/A |
| shYEATS2 #3 | This Study | N/A |
| shYEATS2 #4 | This Study | N/A |
| **Antibodies** | | |
| Anti-YEATS2 | ProteinTech | 24717-1-AP |
| GAPDH | Cell Signaling Technology | 14C10 |
| Anti-β-Actin | Cell Signaling Technology | 12620S |
| Anti-Lamin | Cell Signaling Technology | 2032S |
| Anti-Histone H3 | Abcam | ab1791 |
| 4G10 | EMD Millipore corp | 05-321 |
| Anti-BRD4 | Novus Biological | NBP1-18874 |
| Anti-PanKcr | PTM BioLab | PTM-501 |
| Anti-ZZZ3 | SIGMA | SAB4501106 |
| Anti-H3K27cr | PTM BioLab | PTM-545RM |
| Anti-H3K27ac | PTM BioLab | PTM-116 |
| Anti-cMYC | Cell Signaling Technology | 9402S |
| Anti-rabbit-IgG | ABclonal | AC005 |
| Anti-FLAG | SIGMA | F1804-200UG |
| Anti-mouse-IgG | CST | 7076S |
| **Oligonucleotides** | | |
| Cloning Primers | | |
| Name | | Sequence (5′-3′) |
| NheI_*MYC*-490 F | | CTAGCTAGCTCTGCTCCCTTCTCTTCTCTCA |
| EcoRI_*MYC*-490 R | | CGCGAATTCATAATCATAATCATACTGTTCA |
| NheI_*MYC*-425 F | | CGGCTAGCAGTGAGATGAACCCGGTA |
| XbaI_*MYC*-425 R | | TGCTCTAGATATCAATTATACCTCAATAAATAGGAA |
| ChIP primers | | |
| *MYC*-490-kb enhancer F | | TCTGCTCCCTTCTCTTCTCT |
| *MYC*-490-kb enhancer R | | ACCACCACACTCCATCTTTC |
| *MYC*-425-kb enhancer F | | GAACTGAGGTCGCAAGACAA |
| *MYC*-425-kb enhancer R | | GTAGACCGGAGCTGTTCCTA |
| *MYC* Promoter F | | CAGACACATCTCAGGGCTAAAC |
| *MYC* Promoter R | | TTGGATACCTTCCACCCAGA |
| *MYC* gene body F | | GGGCCTCACACCGAATAAC |
| *MYC* gene body R | | CACCAGACTAGGAAGCAACAA |
| shRNA targeting sequences | | |
| shscramble F | | CCGGCCTAAGGTTAAGTCGCCCTCGCTCGAGCGAGGGCGACTTAACCTTAGGTTTTTG |
| shscramble R | | AATTCAAAAACCTAAGGTTAAGTCGCCCTCGCTCGAGCGAGGGCGACTTAACCTTAGG |

| Reagent/resource | Reference or source | Identifier or catalog number |
|---|---|---|
| sh*MYC*-490eRNA_F #1 | | CCGGAAGGTCAAGTTACTGGAAAGACTGCAGTCTTTCCAGTAACTTGACC TTTTTTTG |
| sh*MYC*-490eRNA_R #1 | | AATTCAAAAAAGGTCAAGTTACTGGAAAGACTGCAGTCTTTCCAGTAAC TTGACCTT |
| sh*MYC*-490eRNA_F #2 | | CCGGTGGTGGTTAATAAAGTGAAACTCGAGTTTCACTTTATTAACCACCATTTTTTTG |
| sh*MYC*-490eRNA_R #2 | | AATTCAAAAATGGTGGTTAATAAAGTGAAACTCGAGTTTCACTTTATTAACCACCAT |
| sh*MYC*-490eRNA_F #3 | | CCGGGATGGAGTGTGGTGGTTAATACTCGAGTATTAACCACCACACTCCATCTTTTTG |
| sh*MYC*-490eRNA_R #3 | | AATTCAAAAAGATGGAGTGTGGTGGTTAATACTCGAGTATTAACCACCACACTCCATC |
| sh*MYC*-490eRNA_F #4 | | CCGGCTTCTCTTCTCTCAGTGAAATCTCGAGATTTCACTGAGAGAAGAGAAGTTTTTG |
| sh*MYC*-490eRNA_R #4 | | AATTCAAAAACTTCTCTTCTCTCAGTGAAATCTCGAGATTTCACTGAGAGAAGAGAAG |
| shYEATS2_F #1 | | CCGG TTCCTTCATCCTAGCTATAAA CTCGAG TTTATAGCTAGGATGAAGGAA TTTTTG |
| shYEATS2_R #1 | | AATTCAAAAA TTCCTTCATCCTAGCTATAAA CTCGAG TTTATAGCTAGGATGAAGGAA |
| shYEATS2_F #2 | | CCGG ATAACAGCAATATGGATATAG CTCGAG CTATATCCATATTGCTGTTATTTTTTG |
| shYEATS2_R #2 | | AATTCAAAAA ATAACAGCAATATGGATATAG CTCGAG CTATATCCATATTGCTGTTAT |
| shYEATS2_F #3 | | CCGG CGTCAGAGTTCAAGTTCATTT CTCGAG AAATGAACTTGAACTCTGACG TTTTTG |
| shYEATS2_R #3 | | AATTCAAAAA CGTCAGAGTTCAAGTTCATTT CTCGAGAAATGAACTTGAACTCTGACG |
| shYEATS2_F #4 | | CCGG AGTACAGGAAGTCCTACAAAC CTCGAG GTTTGTAGGACTTCCTGTACT TTTTTG |
| shYEATS2_R #4 | | AATTCAAAAA AGTACAGGAAGTCCTACAAAC CTCGAGGTTTGTAGGACTTCCTGTACT |
| RT-PCR Primers | | |
| *IL-6* F | | CCAGGAGAAGATTCCAAAGATGTA |
| *IL-6* R | | CGTCGAGGATGTACCGAATTT |
| *IL-1β* F | | CTCTCACCTCTCCTACTCACTT |
| *IL-1β* R | | TCAGAATGTGGGAGCGAATG |
| *MYC*-490-kb eRNA F | | TCTGCTCCCTTCTCTTCTCT |
| *MYC*-490-kb eRNA R | | ACCACCACACTCCATCTTTC |
| *MYC*-425-kb eRNA F | | GAACTGAGGTCGCAAGACAA |
| *MYC*-425-kb eRNA R | | GTAGACCGGAGCTGTTCCTA |
| *MYC* mRNA F | | CCCTCCTACGTTGCGGTCAC |
| *MYC* mRNA R | | GTCCGGGTCGCAGATGAAACT |
| *GAPDH* F | | CAGCCTAGGATCATCAGCAAT |
| *GAPDH* R | | GGTCATGAGTCCTTCCACGA |
| *PMM1* F | | CGCCAGAAAATTGACCCTGAG |
| *PMM1* R | | TTACAGTAGTCAGAGCCGCC |
| *MYC* Intronic primer F | | CCGCATATCGCCTGTGTGAG |
| *MYC* Intronic primer R | | AGTGTCCGTCTCCGGCTGTC |
| Chemicals, enzymes, and other reagents | | |
| DMEM High Glucose | Gibco | 11965118 |
| DMEM low Glucose | Invitrogen | 11885084 |
| FBS | Gibco | 16000044 |
| Pen/Strep | Gibco | 15140122 |
| puromycin | Invitrogen | A1113803 |
| EGF | Gibco | PHG0311L |
| recombinant TNF-α | Abclonal | 30021400 |
| TRIzol reagent | Ambion, Life Technologies | 15596026 |
| Verso cDNA Synthesis Kit | Thermo Fisher Scientific | AB1453B |
| HEPES | Gibco | 15630080 |
| KCL | SIGMA | P4504 |

| Reagent/resource | Reference or source | Identifier or catalog number |
|---|---|---|
| EDTA | SRL | 43272 |
| Protease Inhibitor Cocktail | Invitrogen | 87786 |
| PMSF | SIGMA | 539132 |
| Glycerol | SIGMA | 49767 |
| DTT | Millipore | 1114740005 |
| MgCl$_2$ | SIGMA | 208337 |
| PowerUP SYBR Green PCR Master Mix | Applied Biosystems | A25742 |
| RNasin | Thermo Fisher Scientific | N8080119 |
| NaCl | SIGMA | S9625 |
| Dynabeads™ Protein A | Invitrogen | 10001D |
| Dynabeads™ Protein G | Invitrogen | 10004D |
| NP-40 | MERCK | STS0002 |
| DSG | SIGMA | 80424-50MG-F |
| Glycine | SIGMA | 4840-5KG |
| Triton X-100 | Merck | 112298 |
| Sodium Butyrate | SIGMA | B5887-5G |
| SDS | SIGMA | L6026 |
| Trizma base | SIGMA | T4661 |
| HCl | RANKEM | H0070 |
| Sodium deoxycholate | SIGMA | 30970 |
| Paraformaldehyde (PFA) | SIGMA | P6148 |
| PBS | Gibco | 10010049 |
| lithium chloride (LiCl) | SIGMA | 213233 |
| Bromophenol blue | SIGMA | 114391 |
| Sodium orthovanadate | SIGMA | 450243 |
| M2 Flag beads | SIGMA | M8823 |
| MTT reagent | Sigma Aldrich | M5655 |
| Q5 Site-Directed Mutagenesis Kit | New England Biolabs | E0552S |
| Lipofectamine 3000 | Invitrogen | L3000015 |
| NheI | NEB | R313S |
| EcoRI | NEB | R3101S |
| XbaI | NEB | R0145S |
| AgeI | NEB | R3552L |
| NcoI | NEB | R3193S |
| **Software** | | |
| UCSC genome browser | | https://genome.ucsc.edu/ |
| Adobe Photoshop 7.0 | | |
| HDOCK | | http://hdock.phys.hust.edu.cn/ |
| GraphPad Prism 9.1 | | |
| BioRender | | www.biorender.com |
| ImageJ | National Institutes of Health | |
| RPIseq | | http://pridb.gdcb.iastate.edu/RPISeq |
| FastQC v0.11.7 | | https://www.bioinformatics.babraham.ac.uk/projects/fastqc/ |
| Bowtie2 | | https://bowtie-bio.sourceforge.net/bowtie2/index.shtml |

| Reagent/resource | Reference or source | Identifier or catalog number |
|---|---|---|
| Homer | | http://homer.ucsd.edu/homer/ |
| Trim Galore v0.6.10 | | https://github.com/FelixKrueger/TrimGalore |

HPNE cells were propagated using DMEM (low glucose) medium containing 10% FBS, 1% Pen/Strep, 1.5 µg/ml puromycin (Invitrogen) and 50 ng/ml EGF (Gibco). For experiments involving TNF-α treatment, all specified cells were treated with 20 ng/ml of recombinant TNF-α (Abclonal) for the designated time periods. (0 h—untreated, 24 h—treated) before harvesting for gene expression, UV-RIP, ChIP analysis and other assays.

## Patient samples

We have recruited 7 chronic pancreatitis (CP) (female = 2, male = 5) patients with age 19–45 years (mean age 29.8 years) and 7 non-cancerous (female = 3, male = 4) individuals with age 42–55 years (mean age 48 years). For pancreatic ductal adenocarcinoma (PDAC) patients, we have obtained 5 (female = 3, male = 2) tumor samples and paired adjacent normal tissue. Mean age of the patients was 50.6 years (range: 43–57 years). All the cancer patients were non-diabetic with either moderately or well-differentiated adenocarcinoma, confirmed by H&E staining. For all the above study participants, pancreatic tissues samples were collected by surgical resection from 3 different hospitals in Kolkata (CNCI, SSKM and R.G. Kar) with informed voluntary consent and relevant clinical information (CA19-9, TNM grading, stage, extent of differentiation, smoking habit, alcohol consumption status etc.) at the time of surgery; which are available at Table EV1 for CP and non-cancerous individuals and Table EV2 for PDAC patients. All the patients were treatment-naive at the time of sample collection. The study received approval from the Institutional Ethics Committee (IEC) of SSKM (Inst/IEC/2015/218), CNCI (A-4.311/IG-SG/14/12/2017) and R.G. Kar Medical College and Hospital (IEC/22-01-16).

## Cloning and transfection

The cDNA of human *MYC*-490-kb eRNA and *MYC*-425-kb eRNA was sub-cloned into pcDNA3.1 and YEATS domain (203–329 aa) of YEATS2 protein was sub-cloned in pIRES-neo vector for over-expression experiments. We selected the YEATS domain amino acid sequences from PDB (Protein Data Bank) [https://www.rcsb.org/). Site-directed mutagenesis of the YEATS domain was performed using the Q5 Site-Directed Mutagenesis Kit (New England Biolabs) according to the manufacturer's protocol. The shRNA oligoes against *MYC*-490-kb eRNA and scramble RNA were obtained from Eurofins and cloned into pLKO.1-TRC cloning vector for the knockdown experiment according to the prescribed manufacturer's protocol (Addgene). All the clones were verified through sanger sequencing. The shRNAs transfection was carried out using Lipofectamine 3000 (Invitrogen) according to the manufacturer's protocol into MIAPaCa-2 cells and incubated for 72 h and lysed for subsequent experiments. Primers for Cloning are listed in the Oligonucleotide table.

## Cellular fractionation

MIAPaCa-2 and HPNE cells were treated with 20 ng/ml TNF-α for 24 hr in a six-well culture-based format. After incubation, cells were collected in Buffer A (10 mM HEPES at pH 7.5, 10 mM KCl, 0.1 mM EDTA, 0.1 mM EGTA, 1× Protease Inhibitor Cocktail and 1 mM PMSF) and incubated for 10 min on ice. Then, 0.5% NP-40 was added to the solution, vortexed moderately for 30 s and incubated for 30 s on ice followed by centrifugation at 12,000× g for 1 min at room temperature. The Cytoplasmic Extract was collected. The nuclear pellet was then washed twice with Buffer A (without NP-40) and resuspended the pellet completely in Buffer C (20 mM HEPES at pH 7.5, 420 mM KCl, 0.2 mM EDTA, 1.5 mM MgCl₂, 25% Glycerol, 1 mM DTT, 1× Protease Inhibitor Cocktail and 1 mM PMSF) and vortexed at high speed for 30 s, followed by 2 min incubation on ice, five times. The Nuclear Extract (NE) was collected after centrifugation at 16,000×g for 5 min at 4 °C.

## RNA isolation and qRT-PCR

Total RNA was isolated using TRIzol reagent (Ambion, Life Technologies) from TNF-α treated (24 h) and untreated (0 h, control) pancreatic cell line. cDNA was prepared from the total RNA (250 ng) using Verso cDNA Synthesis Kit (Thermo Fisher Scientific) with random hexamers on MiniAmp Plus thermocycler (Thermo Fisher Scientific). Quantitative real-time PCR (RT-PCR) reactions were performed for each sample in duplicates on CFX96 (C1000 Touch) RT-PCR system (BioRad) using PowerUP SYBR Green PCR Master Mix (Applied Biosystems). The relative expression levels of eRNA and mRNA were determined using the ΔΔCt method, with all expression data normalized to GAPDH, which served as the internal control. The gene expression levels were measured after TNF-α treatment with respect to the levels before TNF-α treatment. All sequences of the primers used for qRT-PCR have been listed in the Oligonucleotide table.

## Nascent RNA isolation and library preparation

1. The nascent RNAs were captured using Click-iT kit (Invitrogen), as described previously (Li et al, 2022). Briefly, after TNF-α (0 h, 24 h) treatment, cells were labeled with EU (0.5 mM), given a pulse for 1 h at 37 °C and then the media was changed and chased for 30 min before completion of 24 h time points.
2. Media was aspirated, cells were washed with 1× PBS and Cellular fractionation was done as before and total RNA was prepared from both the cytosolic and nuclear fractions.
3. Then the nascent RNAs were biotinylated and pulled down by streptavidin beads following the kit protocol. cDNA was prepared on the beads by using a Verso cDNA synthesis kit followed by RT-PCR by SYBR green method.
4. For Nascent RNA sequencing, ribosomal RNA (rRNA) depletion was performed during cDNA library preparation. For rRNA depletion, Streptavidin bead-bound nascent RNA (SVBbnRNA) was hybridized with rRNA-complementary DNA oligonucleotides, followed by RNase H digestion.
5. The rRNA-depleted SVBbnRNA was captured using a magnetic stand, and the supernatant was purified with 2.2× KAPA Pure

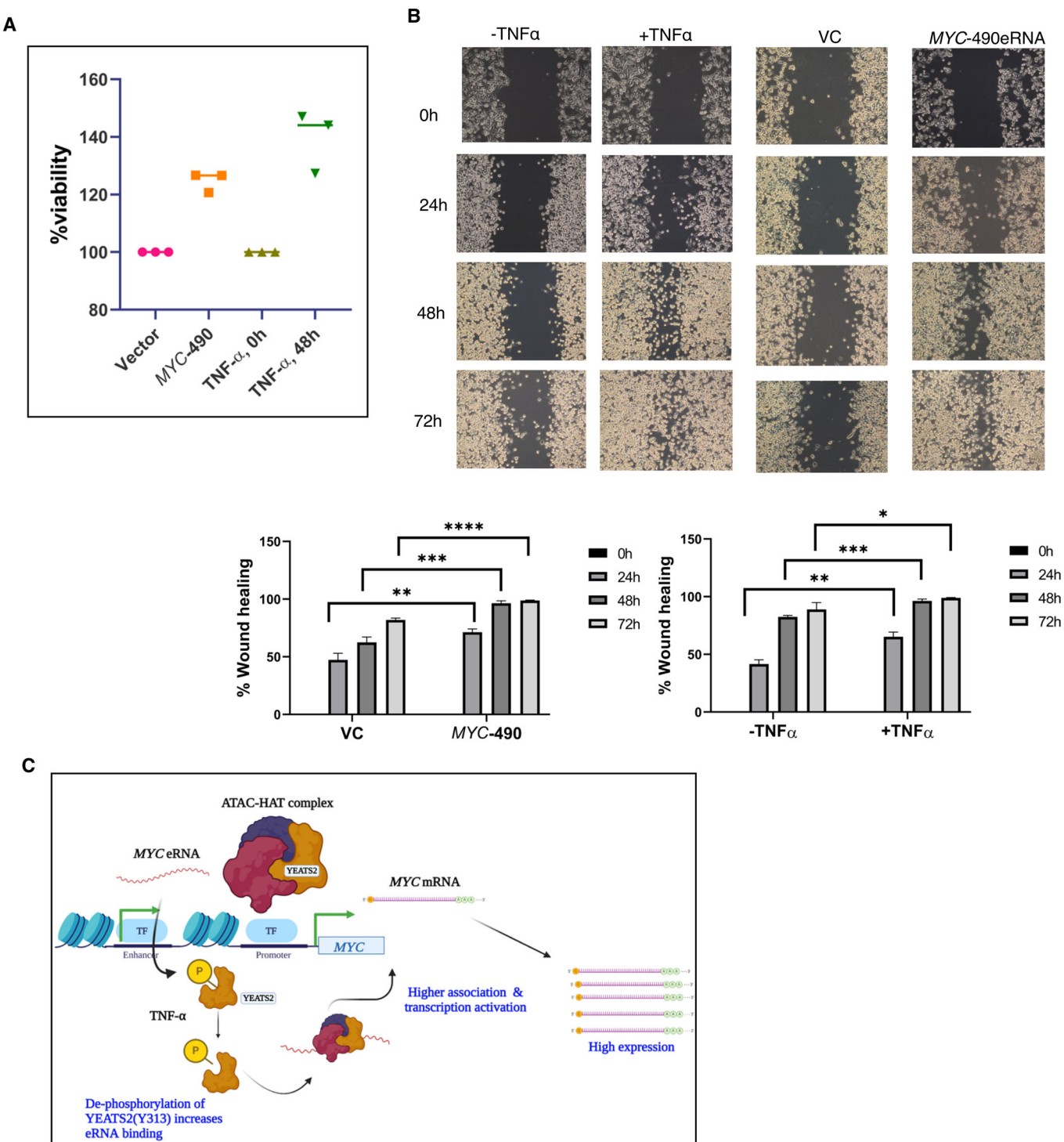

**Figure 5. *MYC* eRNA driven *MYC* gene upregulation leads to cancer cell progression.**

(A) MTT assay was performed from *MYC*-490 eRNA overexpressed condition as well as with TNF-α stimulation in MIAPaCa-2 cells to check cell proliferation ($n = 3$). (B) Wound healing assay revealed that both *MYC*-490-kb eRNA overexpression as well as TNF-α stimulation induce cell migration in MIAPaCa-2 cells. Data are presented as mean ± SD from three independent experiments ($n = 3$). Unpaired $t$ test: **$P = 0.0028$ for 24 h, ***$P = 0.0003$ for 48 h and ****$P = 0.0001$ for 72 h of *MYC*-490 eRNA overexpression, and **$P = 0.0018$ for 24 h, ***$P = 0.0003$ for 48 h and *$P = 0.0448$ for 72 h of TNF-α stimulation. Representative images are shown here from three independent experiments and the dark regions define the areas lacking cells (wound area). (C) Proposed model of *MYC* eRNA-mediated upregulation of *MYC* gene expression in PDAC. Source data are available online for this figure.

Beads before elution in DNase digestion master mix.

6. The eluted fraction was then combined with the rRNA-depleted SVBbnRNA collected from the magnetic separator, followed by DNase treatment to degrade residual rRNA and complementary DNA oligonucleotide duplexes.

7. The resulting supernatant underwent an additional purification step using 2.2× KAPA Pure Beads and was eluted in diluted Fragment, Prime, and Elute buffer, before being mixed with the rRNA-depleted, DNase-treated SVBbnRNA.

Note: Steps 4–7 are novel techniques that we employed to do the Nascent RNA sequencing in ex vivo condition from both the cytosolic and nuclear fractions, thus we can bypass the off-target effects of artificial induction of transcription during GRO-seq study. So, these steps should be followed carefully during the cDNA preparation.

8. Fragmentation of the purified nascent RNA was induced by heat and magnesium.

9. First-strand cDNA synthesis was performed using random priming, followed by combined second-strand synthesis and A-tailing. During these steps, nascent RNA (nRNA) was released from streptavidin DynaBeads, forming cDNA:RNA hybrids.

10. These hybrids were converted into double-stranded cDNA (dscDNA) with dUTP incorporation into the second strand and dAMP addition to the 3' ends of the dscDNA library inserts.

11. Streptavidin DynaBeads were then removed via magnetic separation, and the supernatant containing A-tailed cDNA was collected and ligated with dsDNA adapters containing 3' dTMP overhangs.

12. Limited-cycle high-fidelity PCR amplification was performed, ensuring strand-specific sequencing by preventing the amplification of dUTP-containing strands.

## Library quality control and sequencing

Library quality was assessed using high-sensitivity D1000 Screen-Tape on the Agilent 2200 TapeStation system, and quantification was performed via real-time PCR (QuantStudio7 Flex). Libraries were sequenced on the NovaSeq 6000 platform (Illumina), generating ~50 million paired-end reads of 100 base pairs each.

## Bioinformatic processing

1. Quality control of raw sequencing reads was conducted using FastQC v0.11.7, and adapter trimming was performed with Trim Galore v0.6.10.

2. Reads were aligned to the human genome (hg38) using Bowtie2 (Langmead and Salzberg, 2012) with default parameters, retaining only uniquely aligned reads for downstream analysis.

3. SAM files were then used as input to create tag directories with uniquely aligned reads using Homer (Heinz et al, 2010).

4. Transcripts were called with findPeaks module of the Homer Package with the option "-style groseq".

5. Bigwig files for both strands were generated using the makeBigWig.pl with the option "-strand" as described in the Homer package and visualized in the UCSC genome browser (Perez et al, 2025).

6. The RNA-seq data generated in this study have been deposited in

the Gene Expression Omnibus (GEO) under accession number [GSE288088].

## Immunoblotting and antibodies

Protein samples were isolated from cells by using 1× Passive Lysis Buffer (Promega), 5× SDS-Dye were added to every sample and heated at 95 °C for 5 min. SDS-PAGE was carried out to separate proteins according to their molecular weight and transferred the PVDF membrane. The membrane was incubated overnight with specific primary antibodies and then for 1 h with secondary antibodies before color development. Active bands were detected by ECL (Invitrogen) and visualized by Chemidoc (BioRad). Antibodies used for immunoblotting were sourced in the following manner: anti-YEATS2 (24717-1-AP, ProteinTech, 1:1000), GAPDH (14C10, Cell Signalling Technology, 1:2000), anti-Beta-Actin (12620S, Cell Signalling Technology, 1:2000), anti-Lamin (2032S, Cell Signalling Technology, 1:1000), anti-Histone H3(ab1791, abcam, 1:1000), 4G10 (05-321, EMD Millipore corp, 1:1000), anti-BRD4 (NBP1-18874, Novus Biological, 1:1000).

## Ultraviolet-RNA immunoprecipitation (UV-RIP)

1. UV-RIP assay for the detection of RNA-protein interaction were performed as described earlier (Rahnamoun et al, 2018). MIAPaCa-2 or HPNE cells, treated with TNF-α for 0 or 24 h, were UV-cross-linked (2000 J per cm²) in UVP cross-linker and cells were lysed on ice for 30 min in RIP lysis buffer (25 mM HEPES pH 7.5, 150 mM NaCl, 1.5 mM MgCl₂, 10% glycerol, protease inhibitor cocktails (1:250), 1 mM PMSF, 0.5%NP-40, and RNasin (2–4 U/ml)).

2. Once clear cell lysates were obtained, they were used for IP with 1.0 μg YEATS2 and IgG-antibody overnight at 4 °C.

3. After overnight incubation, antibody-bound samples were incubated with 15 μl Protein G Dynabeads (Invitrogen) and subsequently, the samples were washed three times with RIP lysis buffer, and the RNA was eluted using 200 μl of TRIzol LS reagent (Ambion, Life Technologies).

4. cDNA samples were synthesized as described earlier and analyzed using qRT-PCR with primers listed in the Oligonucleotide table.

## Co-immunoprecipitation

Whole-cell lysates of MIAPaCa-2 and HPNE cell lines were used to perform this assay. Cleared lysates were incubated with specific antibodies (as mentioned before) for overnight and then incubated with Protein G dynabead (Invitrogen) for 4 h. The antibody-bound beads were washed with Wash buffer ((20 mM Tris-HCl, pH 7.5, 200 mM KCl, 5 mM MgCl₂, 1 mM PMSF, 1 mM DTT, 0.5% NP-40) thrice and analyzed by immunoblotting.

## Chromatin immunoprecipitation assay (ChIP-qPCR)

1. ChIP assays were conducted following previously established protocols (Rahnamoun et al, 2018). In brief, MIAPaCa-2 cells, either untreated or treated with 20 ng/ml TNF-α for specified time

points (0 h and 24 h), were sequentially cross-linked with 2 mM DSG (disuccinimidyl glutarate; SIGMA) for 30 min and with 1% formaldehyde for 15 min at room temperature (20–25 °C).

2. The cross-linking was then quenched with 125 mM glycine (Sigma) for 5 min.

3. Cells were lysed in 1 ml of lysis buffer (20 mM Tris-HCl, pH 7.5; 300 mM NaCl; 2 mM EDTA; 0.5% NP-40; 1% Triton X-100; 1 mM PMSF; PICs (1:250); 1 mM sodium butyrate) and incubated on ice for 30 min.

4. The cell suspension was then homogenized in an ice-cold Dounce homogenizer.

5. Nuclear pellets were collected and reconstituted in shearing buffer (1% SDS; 0.5%, 50 mM Tris-HCl, pH 8.1; 10 mM EDTA; 1 mM PMSF; PICs (1:250); 1 mM sodium butyrate).

6. Chromatin was fragmented by sonication three times [15 sec on cycle, 45 sec off cycle, 40% efficiency] to an average size of 200–600 bp using an Ultrasonic Processor (ChromTech).

7. Precleared chromatin was incubated with antibodies overnight at 4 °C for immunoprecipitation, and the resulting immunocomplexes were captured using protein G Dynabeads.

8. A total of 65 μg of sheared chromatin was used for the immunoprecipitation. The immunocomplexes were washed eight times with wash buffer at 4 °C (50 mM HEPES, pH 7.6; 500 mM LiCl; 1 mM EDTA; 1% NP-40; 0.7% sodium deoxycholate; 1 mM PMSF; PICs (1:500); 1 mM sodium butyrate), followed by two washes with 1×TE buffer. The complexes were then eluted in elution buffer (50 mM Tris-HCl, pH 8.0; 10 mM EDTA; 1% SDS) at 65 °C for 30 min, and cross-links were reversed at 65 °C overnight.

9. DNA was purified using the QIAprep Spin Miniprep Kit (Qiagen). qPCR was carried out on an Applied Biosystems CFX96 (C1000 Touch) real-time PCR system using PowerUP SYBR Green PCR Master Mix (Applied Biosystems).

10. The relative quantities of ChIP DNA were measured against input samples. Primers for ChIP-qPCR are detailed in the Oligonucleotide table.

11. The following antibodies were used for ChIP assays: anti-PanKcr (1 μg), anti-ZZZ3 (1 μg), anti-H3K27cr (1 μg), anti-H3K27ac (1 μg), anti-cMYC (1 μg) and anti-IgG (1 μg).

## Protein chromatin immunoprecipitation

1. The protein ChIP-IP experiment for detecting YEATS2-histone interactions in cells was carried out as previously described (Mi et al, 2017). In this experimental procedure, MIAPaCa-2 cells underwent treatment with TNF-α for 0 and 24 h.

2. Following the treatments, the cells were subjected to cross-linking using a 1% paraformaldehyde (PFA) solution for a period of 10 min.

3. The cross-linking reaction was subsequently halted by the addition of 125 mM glycine for 5 min. The cells were then rinsed once with phosphate-buffered saline (PBS).

4. Next, the cells were lysed using 500 μL of Chromatin Immuno-precipitation (ChIP) lysis buffer [50 mM HEPES pH 7.5, 150 mM NaCl, 1 mM EDTA, 1% T × 100, 0.1% sodium deoxycholate, PIC] and this lysis step was carried out for 30 min on ice.

5. Subsequently, the lysed samples were subjected to sonication three times [15 s on cycle, 45 s off cycle, 40% efficiency], and then centrifugation was done at 20,000×g for 20 min.

6. The resulting supernatant, referred to as the "lysate," was further diluted with an equal volume of cell lysis buffer [20 mM Tris pH 8.0, 150 mM NaCl, 1% T × 100, 1 mM EDTA, 0.01% SDS, 1 mM PMSF, PIC].

7. The diluted lysate was then incubated overnight with a 1 μg antibody, which was chosen for its specific binding to the target protein of interest.

8. On the following day, the lysate-antibody mixture was subjected to a 2 h incubation with protein G Dynabeads to facilitate immunoprecipitation.

9. Following immunoprecipitation, the protein G Dynabeads were subjected to a series of washing steps.

10. First, they were washed once with a low salt buffer [20 mM Tris pH 8.0, 150 mM NaCl, 2 mM EDTA, 1% TX100, 0.1% SDS] followed by subsequent washes with a high salt buffer [20 mM Tris pH 8.0, 500 mM NaCl, 2 mM EDTA, 1% TX100, 0.1% SDS] and a lithium chloride (LiCl) buffer [20 mM Tris pH 8.0, 250 mM LiCl, 1 mM EDTA, 1% NP-40, 1% Sodium deoxycholate].

11. Finally, the proteins of interest were eluted from the Dynabeads by incubation with 2× SDS dye [4% SDS, 20% glycerol, 120 mM Tris pH 6.8, 200 mM DTT, 0.1% Bromophenol blue], and this mixture was heated at 95 °C for 30 min.

12. This process allowed for the recovery and analysis of the target proteins for further scientific investigation.

## In vitro phosphatase assay

1. In six-well plates, HEK293T cells were seeded and transfected with Flag-tagged YEATS domain constructs.

2. After 48 h, the cells were lysed using IP lysis buffer (20 mM Tris-HCl, pH 7.5, 200 mM KCl, 5 mM MgCl$_2$, 1 mM sodium orthovanadate, 1 mM DTT), 0.5% Triton X-100, 0.5% NP-40 and 1× EDTA-free protease inhibitor cocktail (SIGMA), and the lysates were incubated with M2 Flag beads (SIGMA) for 2 h at 4 °C to pull down the Flag-tagged YEATS domain.

3. After incubation the bead was washed with wash buffer (20 mM Tris-HCl, pH 7.5, 200 mM KCl, 5 mM MgCl$_2$, 1 mM sodium orthovanadate, 1 mM DTT, 0.5% NP-40) three times.

4. Subsequently, the M2 Flag beads were incubated with MIAPaCa-2 cell lysates, either untreated or treated with TNF-α, at 30 °C for 30 min.

5. The beads were then washed with IP lysis buffer three times. At the final wash, the beads were divided equally for RNA and protein extraction.

6. For immunoblot analysis, the beads were heated at 95 °C after the addition of SDS dye. In addition, TRIzol LS reagent was added to the beads for RNA extraction and subsequent (qRT-PCR) analysis.

## ChIP-seq and ATAC-seq data analysis

H3K27ac and H3K4me1 ChIP-seq data for MIAPaCa-2 cells (GEO ID: GSE64557, Bioproject: PRJNA271302, Sample ID for H3K27ac: GSM1574239, Sample ID for H3K4me1: GSM1574250, Sample ID for input: GSM1574272) and ATAC-seq data for MIAPaCa-2 cells (GEO ID: GSE164974, Bioproject: PRJNA692718, Sample IDs: GSM5024015, GSM5024016, GSM5024017) were downloaded from

the NCBI SRA database in SRA format and converted to FASTQ using SRA toolkit (https://github.com/ncbi/sra-tools/wiki/01.-Downloading-SRA-Toolkit). QC of the reads was done by FastQC v0.11.7. Adapter trimming was performed by Trim Galore version 0.6.10. Reads were mapped to the hg38 genome using bowtie2 with default parameters. SAM files were then used as input to create tag directories with uniquely aligned reads using Homer. For ChIP-seq analysis, peaks relative to input were called using the findPeaks module of the Homer Package with the option "-style histone" whereas for ATAC-seq analysis option "-style dnase" was used. Bigwig files were generated using the makeBigWig.pl and visualized in the UCSC genome browser.

### Interaction analysis of YEATS2 protein and *MYC* eRNA

We employed the RPIseq (http://pridb.gdcb.iastate.edu/RPISeq) software and the HDOCK (http://hdock.phys.hust.edu.cn/) web server to analyze the interaction between the YEATS2 protein and *MYC* eRNA. Initially, a comprehensive list of histone reader molecules (Yun et al, 2011; Li, 2012; Mi et al, 2017) (Table EV3) was compiled and uploaded into the RPIseq software along with the sequence of *MYC* eRNA. The RPIseq software utilizes Random Forest (RF) and Support Vector Machine (SVM) classifiers for the prediction of RNA-protein interactions, generating corresponding interaction scores (Muppirala et al, 2011).

Subsequently, we used the HDOCK web server to further evaluate the RNA-Protein interaction. HDOCK's scoring function is based on a linear combination of various energy terms and the buried surface area, which provides a quantitative assessment of the interaction (Yan et al, 2017). The results from both RPIseq and HDOCK provided insights into the potential interaction between YEATS2 protein and *MYC* eRNA.

### Statistical analyses

Experimental data were typically presented as means ± standard deviation unless otherwise indicated. Statistical significance was assessed using a two-tailed unpaired $t$ test between two experimental groups, with a threshold of $P < 0.05$ for significance. Significance levels are represented as follows: $*P < 0.05$; $**P < 0.01$; $***P < 0.001$; $****P < 0.0001$. No statistical methods were employed to predetermine the sample size.

### Survival plot analysis

Publicly available data from TCGA pancreatic adenocarcinoma (PAAD) cohort and the Gene Expression Omnibus (GEO) dataset GSE-62452 was utilized for the survival analysis of patients with pancreatic tumor. The expression levels of the *MYC* gene were extracted from those datasets. Patients were stratified into two groups based on the median *MYC* gene expression: high expression and low expression. Overall survival (OS) between these two groups was compared. Kaplan–Meier survival analysis was conducted to assess the variation in overall survival among patients with high versus low *MYC* expression. The log-rank test was used to determine the statistical significance of the survival differences. Survival curves were plotted using the ggsurvplot function from the survminer package [https://CRAN.R-project.org/package=survminer] in R Version 4.3.2, which leverages the ggplot2 package for graphical representation.

### Cell proliferation assay

The viability of MIAPaCa-2 in the presence and absence of TNF-α and *MYC*-490-kb eRNA was assessed using a cell proliferation assay (3-(4,5-dimethyl-2-thiazolyl)-2,5-diphenyl-tetrazolium bromide- MTT). A total of $10^4$ numbers of cells were seeded onto 96-well culture plates. To achieve this, cells were incubated with MTT reagent (5 mg/ml, Sigma Aldrich, USA) for 4 h at 37 °C in a 5% $CO_2$ incubator until a purple precipitate developed. Three wells containing only culture medium and MTT were used as blank controls. The optical density was measured at 560 nm using (Infinite M Plex, TECAN).

### Scratch wound healing assay

Cell migration was performed using a scratch wound healing assay as described previously (Bahar et al, 2020). In brief, MIAPaCa-2 cells were cultured in six-well plates in the presence and absence of TNF-α or transfected with *MYC*-490-kb eRNA. Cells were rinsed with PBS to eliminate any debris. Images were captured at 0, 24 h, 48 h, and 72 h after wounding. The gap distance can be quantitatively assessed using software like ImageJ. (National Institutes of Health). The formula used to calculate the percentage of wound healing area is given below. **% of Wound healing = ( wound width at time zero – wound width at the indicated time)/ wound width at time zero *100.**

## Data availability

The FASTQ file for the Nascent RNA sequencing has been uploaded to the Gene Expression Omnibus (GEO) under accession number GSE288088. The data can be accessed via: https://www.ncbi.nlm.nih.gov/geo/query/acc.cgi?acc=GSE288088.

The source data of this paper are collected in the following database record: biostudies:S-SCDT-10_1038-S44319-025-00446-0.

## Peer review information

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

## Acknowledgements

The authors would like to thank Dr. Sagar Sengupta for critical and helpful comments on the manuscript. The authors thank the National Genomics Core of BRIC-NIBMG, especially Dr. Subrata Patra and Dr. Tithi Pal for helping in standardizing the Nascent RNA-sequencing methodology. AM, SG, and NKB would like to thank Department of Biotechnology, India for providing funding. JR would like to thank University Grant Commission, India for providing fellowship. AK would like to thank Counsil of Scientific and Industrial Research, India for providing fellowship. SC would like to thank the Department of Biotechnology, India for providing fellowship.

## Author contributions

**Jayita Roy**: Data curation; Formal analysis; Validation; Investigation; Visualization; Methodology; Writing—original draft; Writing—review and editing. **Aniket Kumar**: Data curation; Software; Formal analysis; Investigation; Methodology. **Shouvik Chakravarty**: Software; Methodology. **Nidhan K Biswas**: Software; Formal analysis; Supervision. **Srikanta Goswami**: Resources; Supervision; Funding acquisition; Project administration. **Anup Mazumder**: Conceptualization; Data curation; Supervision; Funding acquisition; Investigation; Visualization; Writing—original draft; Project administration; Writing—review and editing.

Source data underlying figure panels in this paper may have individual authorship assigned. Where available, figure panel/source data authorship is listed in the following database record: biostudies:S-SCDT-10_1038-S44319-025-00446-0.

## Disclosure and competing interests statement

The authors declare no competing interests.

# Expanded View Figures

**Figure EV1.   Induction of chronic inflammation in mammalian cell lines.**

(A) Survival plot analysis depicting the impact of *MYC* amplification status on the overall survival of patients in the GSE-62452 cohort. Through 24 h treatment of TNF-α, a chronic inflammatory condition was mimicked in (B) MIAPaCa-2, data are presented as mean ± SD from three independent experiments ($n = 3$). Statistical significance was determined using an unpaired two-tailed *t* test: ***$P = 0.0003$ and ***$P = 0.0002$ for IL-6 and IL-1β respectively and (C) AsPC-1, data are presented as mean ± SD from three independent experiments ($n = 3$). Statistical significance was determined using an unpaired two-tailed *t* test: ***$P = 0.00015$ and ****$P = < 0.0001$ for IL-6 and IL-1β respectively. (D) HPNE cell lines which has been shown by significant increase in the expression level of pro-inflammatory cytokines IL-6 and IL-1β. Data are presented as mean ± SD from three independent experiments ($n = 3$). Statistical significance was determined using an unpaired two-tailed *t* test: ***$P = 0.0008$ and ***$P = 0.0006$ for IL-6 and IL-1β respectively. (E) Validation of *MYC* eRNA and *MYC* mRNA expression was performed using RT-PCR in TNF-α stimulated HCT-116 cells for 0 h and 24 h, data are presented as mean ± SD from three independent experiments ($n = 3$). Statistical significance was determined using an unpaired two-tailed *t* test: ****$P = < 0.0001$, **$P = 0.0082$ and ****$P = 0.0001$ for *MYC*-490, *MYC*-425, and *MYC* mRNA respectively and the expression level of pro-inflammatory cytokines IL-6 and IL-1β **$P = 0.0015$, ***$P = 0.0008$ respectively was measured in those conditions.

                                                       

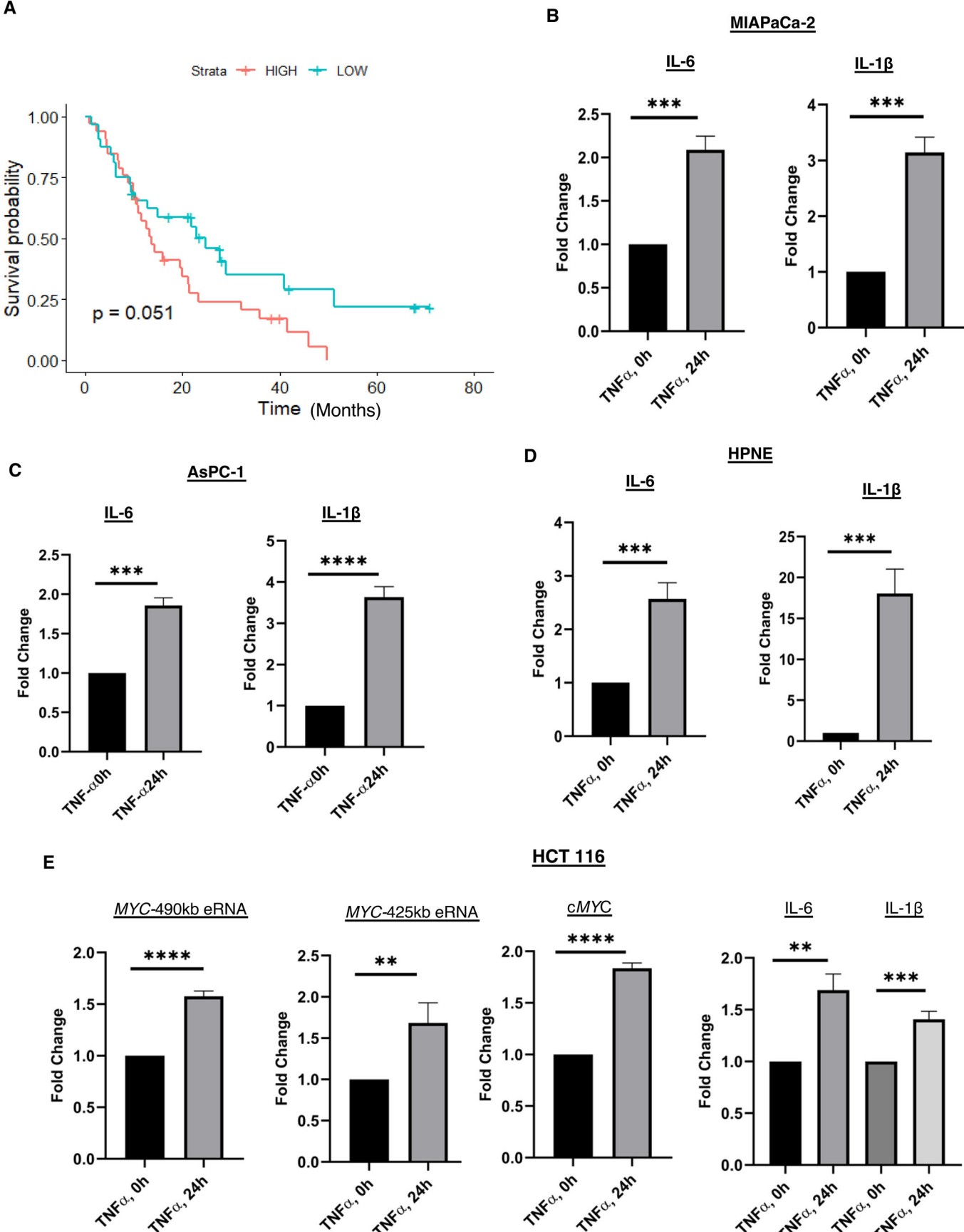

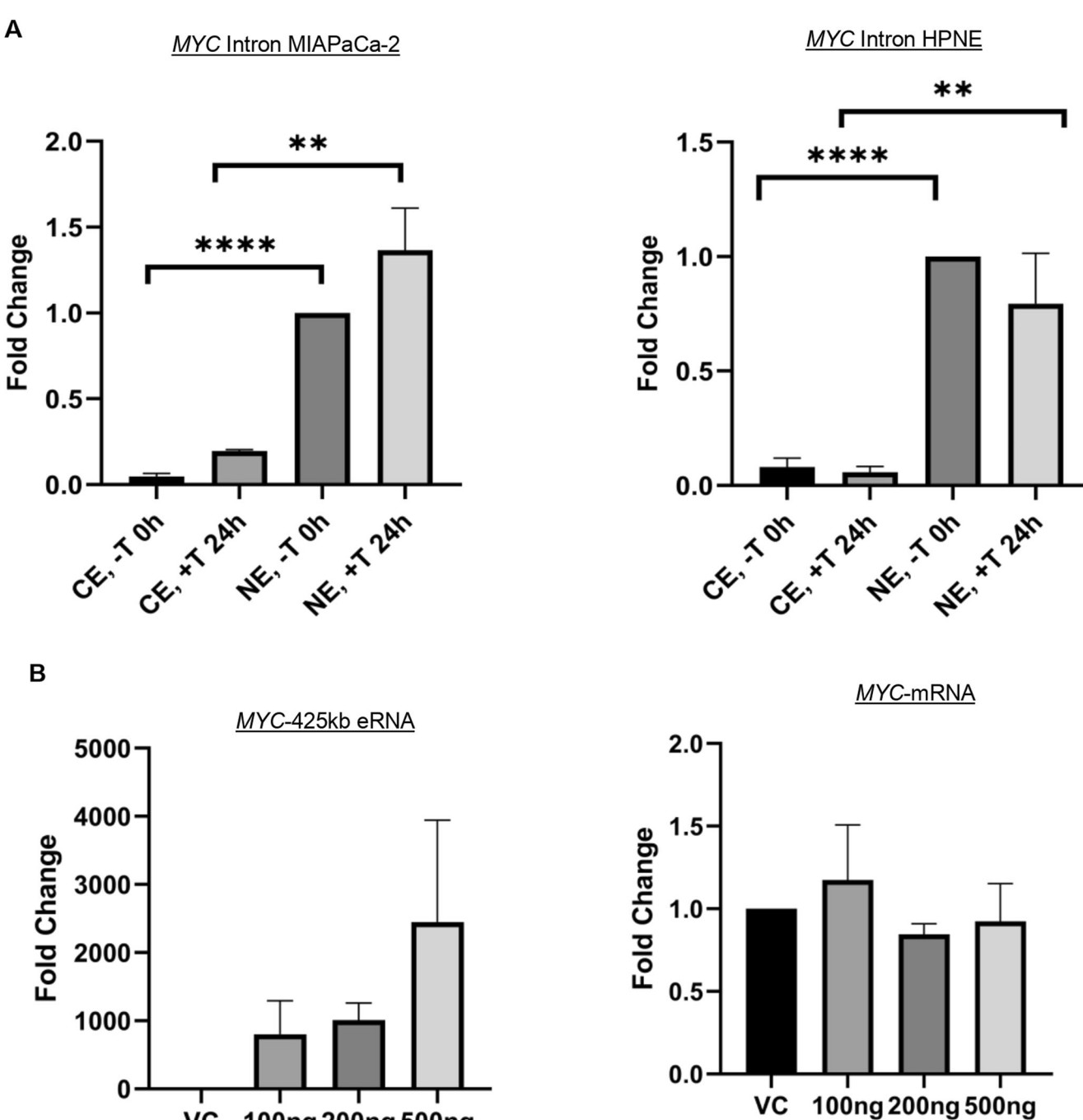

**Figure EV2. Analysis of *MYC* Intron from fractions as well as *MYC*-425-kb eRNA driven *MYC* mRNA expression by RT-PCR.**

(A) *MYC* intron RNA level was checked by qRT-PCR using intron specific primer from cytosolic and nuclear fraction of MIAPaCa-2 (left panel). Data are presented as mean ± SD from three independent experiments (n = 3). Statistical significance was determined using an unpaired two-tailed t test: ****P = < 0.0001 and **P = 0.0012 for TNF-α 0 h and 24 h respectively and HPNE cells (right panel), data are presented as mean ± SD from three independent experiments (n = 3). Statistical significance was determined using an unpaired two-tailed t test: ****P = < 0.0001 and **P = 0.0045 for TNF-α 0 h and 24 h respectively. (B) MIAPaCa-2 cells were transfected with plasmid expressing *MYC*-425-kb eRNA (left panel), at increasing gradient of concentration for 48 h and *MYC* mRNA expression (right panel) were checked by RT-PCR. The data represented the mean and s.e.m. of n = 3 independent experiments.

**A**

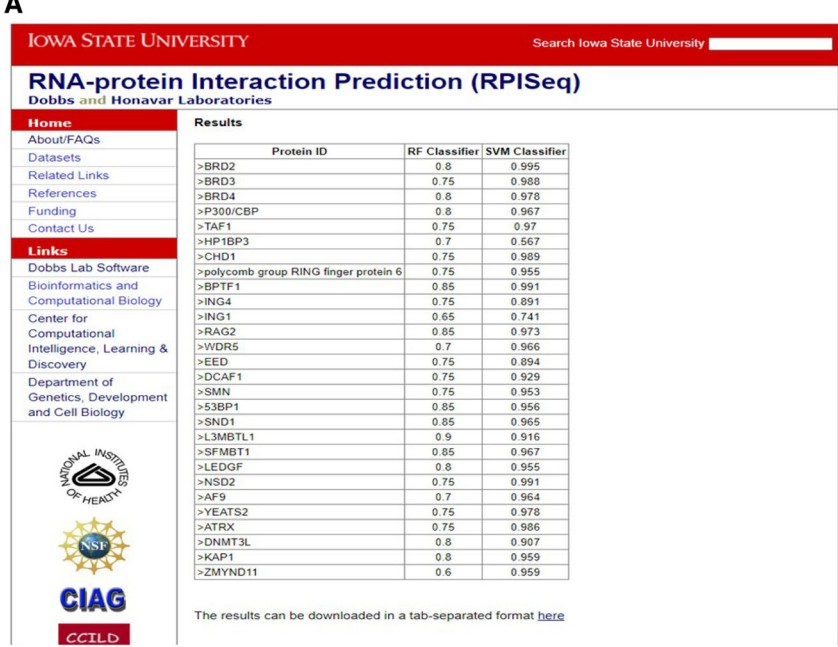

**B**

| Protein | Hdock score |
|---|---|
| BRD2 | 0.9863 |
| BRD3 | 0.9764 |
| BRD4 | 0.9745 |
| P300/CBP | 0.9571 |
| TAF1 | 0.9671 |
| HP1BP3 | 0.8798 |
| CHD1 | 0.9674 |
| PCGF6 | 0.9250 |
| BPTF1 | 0.9437 |
| ING4 | 0.9072 |
| ING1 | 0.9340 |
| RAG2 | 0.8752 |
| WDR5 | 0.9300 |
| EED | 0.9840 |
| DCAF1 | 0.8880 |
| SMN | 0.8728 |
| 53BP1 | 0.9776 |
| SND1 | 0.9546 |
| L3MBTL1 | 0.9810 |
| SFMBT1 | 0.9589 |
| LEDGF | 0.9187 |
| NSD2 | 0.9462 |
| AF9 | 0.9575 |
| YEATS2 | 0.9758 |
| ATRX | 0.9737 |
| DNMT3L | 0.9697 |
| KAP1 | 0.9723 |
| ZMYND11 | 0.9638 |

**C** *MYC*-490kb eRNA

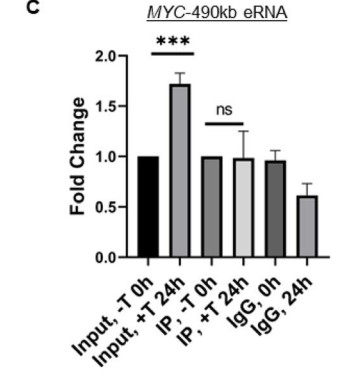

**D** *MYC*-425kb eRNA

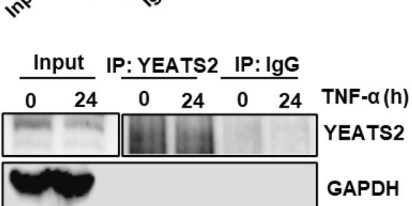

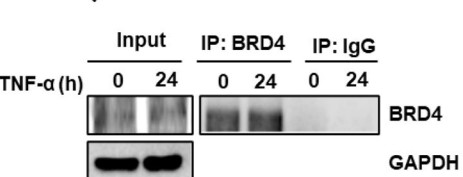

**E**

YEATS2 PROTEIN AA Sequence HUMAN:

MSGIKRTIKETDPDYEDVSVALPNKRHKAIENSARDAAVQKIETIIKEQFALEMKNKEHEIEVIDQRLIEARRMMDKL
RACIVANYYASAGLLKVSEGSKTCDTMVFNHPAIKKFLESPSRSSSPANQRAETPSANHSESDSLSQHNDFLSDKDN
NSNMDIEERLSNNMEQRPSRNTGRDTSRITGSHKTEQRNADLTDETSRLFVKKTIVVGNVSKYIPPDKREENDQST
HKWMVYVRGSRREPSINHFVKKVWFFLHPSYKPNDLVEVREPPFHLTRRGWGEFPVRVQVHFKDSQNKRIDIIHN
LKLDRTYTGLQTLGAETVVDVELHRHSLGEDCIYPQSSESDISDAPPSLPLTIPAPVKASSPIKQSHEPVPDTSVEKAGF
PASTEAERHTPFVALPSSLERTPTKMTTSQKVTFCSHGNSAFQPIASSCKIVPQSQVPNPESPGKSFQPITMSCKIVSG
SPISTPSPSPLPRTPTSTPVHVKQGTAGSVINNPYVIMDKQPGQVIGATTPSTGSPTNKISTASQVSQGTGSPVPKIHG
SSFVTSTVKQEDSLFASMPPLCPIGSHPKVQSPKPITGGLGAFTKVIIKQEPGEAPHVPATGAASQSPLPQYVTVKGG
HMIAVSPQKQVITPGEGIAQSAKVQPSKVVGVPVGSALPSTVKQAVAISGGQILVAKASSSVSKAVGPKQVVTQGVA
KAIVSGGGGTIVAQPVQTLTKAQVTAAGPQKSGSGSQGSVMATLQLPATNLANLANLPPGTKLYLTTNSKNPSGKGKLL
LIPQGAILRATNNANLQSGSAASGGSGAGGGGGGGGGGGGSGSGGGGSTGGGGGTAGGGTQSTAGPGGISQHLT
YTTSVILKQTPQGTFLVGQPSPQTSGKQLTTGSVVQGTLGVSTSSAQGGQTLKVISGQKTTLFTQAAHGGQASLMKIS
DSTLKTVPATSQLSKPGTTMLRVAGGVITTATSPAVALSANGPAQQSEGMAPVSSSTVSSVTKTSGQQQVCVSQATY
GTCKAATPTVVSATSLVPTPNPISGKATVSGLLKIHSSQSSPQQAVLTIPSQLKPLSVNTSGGVQTILMPVNKVVQSFS
TSKPPAILPVAAPTPVVPSSAPAAVAVKVKTEPETPGPSCLSQEGQTAVKTEESSELGNYVIKIDHLETIQQLLTAVVKKIP
LITAKSEDASCFSAKSVEQYYGWNIGKRRAAEWQRAMTMRKVLQEILEKNPRFHHLTPLKTKHIAHWCRCHGYTPP
DPESLRNDGDSIEDVLTQIDSEPECPSSFSSADNLCRKLEDLQQFQKREPENEEEVDILSLSEPVKINIKKEQEEKQEEV
KFYLPPTPGSEFIGDVTQKIGITLQPVALHRNVYASVVEDMILKATEQLVNDILRQALAVGYQTASHNRIPKEITVSNIH
QAICNIPFLDFLTNKHMGILNEDQ

**F**

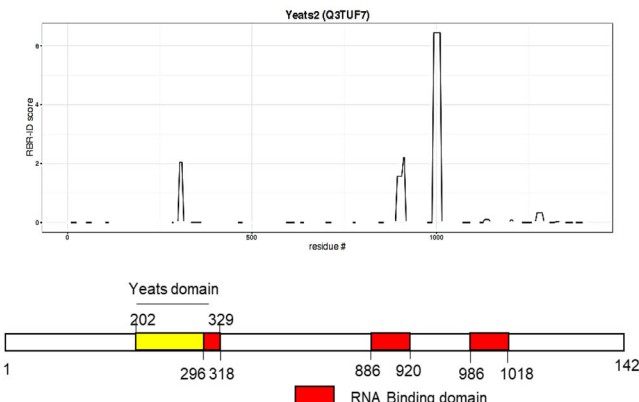

**Figure EV3.  *MYC*-490-kb eRNA interacts with YEATS2, a component of ATAC-HAT complex.**

(A) Analysis by RPIseq software to check the interaction between *MYC*-490-kb enhancer RNA and different histone reader molecules. (B) Analysis by HDOCK web server tool to check the interaction between *MYC*-490-kb enhancer RNA and different histone reader molecules. (C) UV-RIP experiment was performed in MIAPaCa-2 cells in TNF-α stimulated condition and BRD4 associated *MYC*-490 eRNA was checked by qRT-PCR. Data are presented as mean ± SD from three independent experiments ($n = 3$). Statistical significance was determined using an unpaired two-tailed $t$ test: ***$P = 0.0003$ for Input and ns for IP. Western blot was performed to check successful pulldown of BRD4 protein. GAPDH served as loading control. (D) UV-RIP experiment was performed in MIAPaCa-2 cells in TNF-α stimulated condition and YEATS2 associated *MYC*-425 eRNA was checked by qRT-PCR. Data are presented as mean ± SD from three independent experiments ($n = 3$). Statistical significance was determined using an unpaired two-tailed $t$ test: **$P = 0.0037$ for Input. Western blot was performed to check successful pulldown of YEATS2 protein. GAPDH served as loading control. (E) Amino acid sequence of YEATS2 showing the Tyr residue (Red color) throughout the protein. The YEATS domain was marked with yellow color, and the three RNA binding domains were marked with purple color. (F) The three RNA binding motifs as shown in RBR-ID from Bonasio's lab (upper panel). A schematic of YEATS2 protein showing the overlapping region of YEATS domain with the 1st RNA binding motif (lower panel).

                                          

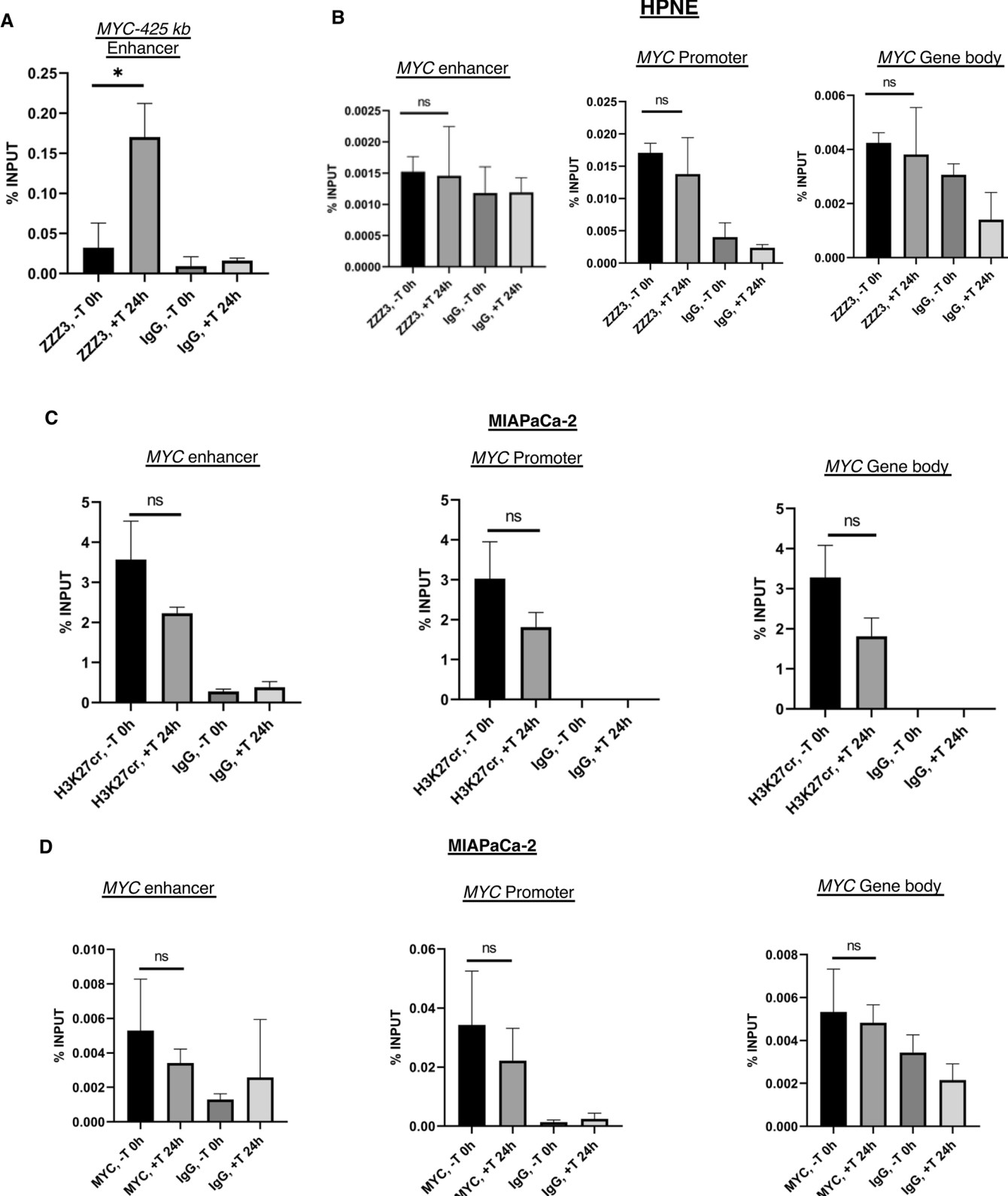

◀ **Figure EV4. Neither MYC occupancy nor H3K27cr level was increased in *MYC* promoter/enhancer region with TNF stimulation.**

(A) ChIP-qPCR analysis was performed using ZZZ3 antibody to check YEATS2-containing ATAC complex occupancy in 24 h TNF-α stimulated condition at *MYC*-425-kb enhancer region. Data are presented as mean ± SD from three independent experiments ($n = 3$). Statistical significance was determined using an unpaired two-tailed *t* test: *$P = 0.0146$ for Input. (B) There was no significant change in YEATS2 occupancy (measured by ZZZ3 antibody) at promoter region or enhancer region (490 kb upstream) as well as gene body of *MYC* gene between 0 h and 24 h of TNF-α treatment in HPNE cell line. (C) ChIP-qPCR study was performed using H3K27cr (crotonylation) antibody from TNF-α treated MIAPaCa-2 cells in both enhancer and promoter region as well as gene body. (D) Similar ChIP-qPCR was done with MYC antibody to assess MYC TF occupancy in MYC promoter and/or enhancer and also in gene body of 0 h and 24 h of TNF-α treated MIAPaCa-2 cells. The data represented the mean and s.e.m. of $n = 3$ independent experiments. Statistical significance was determined by a two-tailed Student's *t* test. ns, not significant.

