## [Peer Review File · EMBO Reports]

Dynamic interaction of MYC enhancer RNA with YEATS2 protein regulates MYC gene transcription in pancreatic cancer

Anup Mazumder, Jayita Roy, Aniket Kumar, Shouvik Chakravarty, Nidhan Biswas, and Srikanta Goswami

Corresponding author(s): Anup Mazumder (am7@nibmg.ac.in) , Srikanta Goswami (sg1@nibmg.ac.in)

Review Timeline:

Submission Date:	19th Aug 24
Editorial Decision:	25th Sep 24
Appeal Received:	14th Feb 25
Editorial Decision:	19th Mar 25
Revision Received:	26th Mar 25
Accepted:	31st Mar 25

Editor: Esther Schnapp

Transaction Report:

Dear Dr. Mazumder,

Thank you for the submission of your manuscript to EMBO reports. We have now received the enclosed reports from 2 referees and given that they are in fair agreement, I am making a decision on your study now, in order to save you from unnecessary loss of time.

As you will see, none of the referees are very positive about the manuscript, and both ask for many more data that would be required to substantiate the claims.

Both referees also rate the technical quality of the manuscript "low/unacceptable" on the summary evaluation sheets returned with their reports.

Given these opinions and the fact that EMBO reports can only invite revision of papers that receive enthusiastic support from the referees, I am afraid that we cannot offer to publish your manuscript.

I am sorry to disappoint you on this occasion, and hope that the referee comments will be helpful in your continued work in this area.

Yours sincerely

Referee #1:

In this study Roy et al identify a YEATS2-MYC-490 kb eRNA interaction which they suggest is critical for MYC hyperactivation in PDAC. While study provides and potentially interesting and novel example of an oncogenic eRNA, I found that the mechanistic data was incomplete and lacking in several critical controls. Thus, I am not convinced that the major manuscript is true. Several additional control experiments are essential before this work should appear in a published manuscript.

Major points:

1. Please add additional shRNAs for eRNA knockdown. For a study like this, at least five independent shRNAs should be employed to ensure that off-target targets are not leading the phenotypes observed.
2. The authors should evaluate whether the MYC eRNA is enriched in a YEATS2 IP relative to control IgG. Controls of other eRNAs should be included that address sequence specificity.
3. Bold claims are made by the authors about tyrosine phosphorylation regulating the eRNA:YEATS2 complex, yet experimental evidence directly shows this. The authors should compare wild-type YEATS2 to Tyr mutant in pulldown experiments with the eRNAs.
4. Perform ChIP-qPCR with YEATS2 +/- eRNA knockdown and comparing wild-type YEATS2 to the tyr mutant.
5. In Fig 2D, it is strange that 100 ng MYC-490 eRNA has no effect on MYC expression, 200 ng massively upregulates MYC expression, and 300 ng does not significantly increase MYC expression. To improve the confidence in this assay, the authors should treat cells with intermediate concentrations of MYC-490 eRNA (150 ng, 175 ng, 225 ng, 250 ng) to evaluate if these concentrations similarly upregulate MYC expression in their system.
6. To support the main claim of this study, it is critical that the authors knockdown/knockout YEATS2 and evaluate whether this alters MYC transcription in their pancreatic cancer models.

Referee #2:

The manuscript by Roy et al reports that MYC transcription in pancreatic tumor cells is regulated in part by MYC enhancer RNAs (eRNAs), especially when treated with TNF-. The authors also show MYC-490 kb eRNA is elevated in chronic pancreatitis

patients. The main thrust of this study was to examine how an eRNA (MYC-490 kb eRNA) influences MYC transcription. Overall, this study provides some new and interesting insights into the roles of MYC eRNAs by linking them to YEATS2 and TNF-elevated MYC transcription. The authors also report that tyrosine phosphorylation of YEATS2 is decreased by TNF-, which enhances the ability of MYC-490 kb eRNA to bind to YEATS2. However, as discussed below, the story is incomplete, and several issues are not adequately addressed.

Major Concerns:

1. This study refers to MYC-490 kb eRNA in the Results section. However, in the Discussion, the authors refer to it as MYC-490 kb eRNAs in at least one instance. Also, the authors show that MYC-425 kb eRNA as well as MYC-490 kb eRNA is upregulated when MIAPaCa-2 cells were treated with TNF-. MYC-490 kb eRNA and MYC-425 kb eRNA are each likely to be a group of eRNAs expressed from their respective enhancer regions. The authors need to clarify this issue for the reader. If the authors believe that MYC-490 kb eRNA is a single eRNA and MYC-425 kb eRNA is a single eRNA, what are the sequences? Also, super enhancers comprise multiple enhancers distributed in some cases over many kb. For example, the upstream MYC super enhancer in colorectal tumor cells has been reported to comprise at least five MYC enhancers distributed over 40 to 50 kb. Are there multiple MYC eRNAs expressed from multiple MYC enhancers that are functional in TNF- treated MIAPaCa-2 cells? In this regard, the authors identified the MYC super enhancer from data acquired in SW480 colorectal cells. A detailed discussion of MYC super enhancers is needed to provide the reader with a better perspective on MYC super enhancers.
2. The authors report that transfection of cells with an shRNA to knockdown MYC-490 kb eRNA reduced both MYC-490 kb eRNA and MYC mRNA to the same extent (Fig 2C). This finding raises two fundamental questions. First, this should only occur if there is a single key MYC eRNA, MYC-490 kb eRNA, that controls MYC transcription. While it may be the case, this seems highly unlikely for reasons discussed above (multiple MYC enhancers and multiple eRNAs). Also, about a 40-fold increase in MYC-490 kb eRNA (after transfection) led to only a 60 or 70% increase in MYC mRNA (Fig 2D). How do the authors explain these differing results. Also, the authors need to show how the transfection of the scrambled shRNAs effected on the levels of MYC-490 kb eRNA and MYC mRNA. Second, the authors report that they transfected the cells with Lipofectamine 3000. What was the transfection efficiency of the MIAPaCa-2 cells with the plasmids used in this study? PDAC cells transfect poorly, usually below 20%. This is key to interpreting the results of their transfection experiments.
3. The authors should check by ChIP-qPCR, the association of the ATAC complex with the MYC-425 enhancer region
4. A significant limitation of this study is that it reports the results with only one PDAC cell line. Are their findings generally true for PDAC cells or specific to MIAPaCa-2 cells? It would be helpful to know if any of the major findings occur in other PDAC cell lines. For example, does TNF- increase the expression of MYC-490 kb eRNA and MYC-425 kb eRNA in other PDAC cell lines? Also does TNF- increase the association of the ATAC complex with the MYC enhancer in other PDAC cell lines?
5. In the discussion, the authors argue that "it is possible that in chronic inflammatory condition, the MYC eRNA confers the specificity of YEAT2 towards crotonyl-lysine binding over acetyl-lysine binding." However, the authors probed for H3K27cr and found no evidence for it. Thus, the authors should omit this commentary until they have data to support it.
6. The effect of MYC-490 kb eRNA on proliferation is very small or non-existent. The difference between the vector control and the MYC-490 kb eRNA transfected cells does not appear to be statistically significant.

Minor Concerns:

1. A better/additional reference regarding MYC enhancers activated in tumor cells is Dave et al eLife 6, e23382 (2017).
2. The authors should provide the reference for the GRO-Seq data mentioned in their study - reference 32?
3. The data in Fig S1E is not mentioned in the manuscript. Either discuss the data or omit it from the manuscript.
4. The decreased expression of MYC-490 kb eRNA reported in Fig 2C left panel and MYC mRNA in Fig 2C right panel is not nearly 50 or 40 to 50%, respectively.
5. Why did the authors select YEATS2 for their ChIP studies? Many other proteins, including p300 are predicted to bind MYC-490 kb eRNA, and p300 has been shown to associate with the MYC super enhancer.
6. The authors refer to H3 in Fig 4E. Was the H3K27ac antibody used? If so, this should be spelled out.
7. The font of many of the figures is too small to read.
8. CBP/p300 function primarily as histone writers not as histone readers.
9. The authors claim that some of their data "proves" their hypothesis. The data that they are referring to is consistent with their hypothesis. You can test a hypothesis, but not prove a hypothesis.

** As a service to authors, EMBO Press provides authors with the ability to transfer a manuscript that one journal cannot offer to publish to another journal, without the author having to upload the manuscript data again. To transfer your manuscript to another EMBO Press journal using this service, please click on

Link Not Available

BRIC
a DBT Organization

राष्ट्रीय जैवचिकित्सा
जीनोमिक्स संस्थान
National Institute of
Biomedical Genomics

Date: 14th February 2025

Dear Dr. Schnapp,

Please find enclosed our manuscript entitled “**Dynamic interaction of *MYC* enhancer RNA with YEATS2 protein regulates *MYC* gene transcription in pancreatic cancer**” by Roy J. *et.al.* for publication into your esteemed journal.

We submitted our manuscript earlier on 16th August 2024 with the manuscript ID: **EMBOR-2024-60223V1**. Here we have identified novel *MYC* enhancer RNA (eRNA) is regulating *MYC* gene transcription in pancreatic cancer cells. Further, we have shown that *MYC* eRNA is dynamically interacting with YEATS2 protein and augmenting the association of YEATS2-containing ATAC complex to *MYC* promoter and enhancer regions to induce *MYC* gene transcription and thus helps in cancer progression. Our manuscript has identified that a TNF- α driven Tyrosine dephosphorylation event at YEATS domain of YEATS2 protein is regulating the eRNA-protein interaction in a cancer cell specific manner.

It was reviewed one time but rejected for publication. But most of the reviewers' comments were addressable. I had an email conversation with you where you said that I can do a new submission along with the point-to point answers to the reviewers' concerns. Now, we have addressed all the reviewers' concerns except one. In the revamped manuscript:

1. We have included the nascent RNA-sequencing data from TNF- α treated MIAPaCa-2 cells to identify the *MYC* eRNAs that are regulating *MYC* gene transcription in a very precise and convincing way. We have taken a useful approach to capture the nascent RNA directly from the cell thus overcoming any artefact that could be generated due to artificial induction of transcription during GRO-seq analysis.
2. We have added the key experimental data done in another pancreatic cancer cell – AsPC-1 to strengthen our claim.
3. We have knocked down YEATS2 and showed less *MYC* mRNA expression in those condition supporting our claim that YEATS2 plays a significant role in *MYC* gene transcription in pancreatic cancer.
4. We have included a greater number of shRNA molecules to knock-down *MYC* eRNA to nullify any off-target effect.
5. Only we could not over-express the Tyr mutant YEATS2 protein in MIAPaCa-2 cells due to poor transfection efficiency of pancreatic cancer cell in general. However, we have done additional experiments to support our claim regarding that issue.

We confirm that this manuscript has not been published elsewhere or is not under consideration by another journal. All contributing authors have approved the manuscript and agreed with its submission to EMBO Reports. We sincerely hope that you will find this revamped manuscript more interesting and hence acceptable for publication. Looking forward to hearing positive results. Thanking you,

Anup Mazumder

Anup Mazumder, *Ph.D.*

BRIC-National Institute of Biomedical Genomics,

Kalyani, West Bengal, India. Pin-741251.

Email: am7@nibmg.ac.in

Referee #1:

In this study Roy et al identify a YEATS2-MYC-490 kb eRNA interaction which they suggest is critical for MYC hyperactivation in PDAC. While study provides and potentially interesting and novel example of an oncogenic eRNA, I found that the mechanistic data was incomplete and lacking in several critical controls. Thus, I am not convinced that the major manuscript is true. Several additional control experiments are essential before this work should appear in a published manuscript.

Major points:

1. Please add additional shRNAs for eRNA knockdown. For a study like this, at least five independent shRNAs should be employed to ensure that off-target targets are not leading the phenotypes observed.

We thank the reviewer for the concern. We showed data for two different shRNA against MYC-490 kb eRNA in the manuscript. Further, we have performed experiment with three new shRNA against the MYC-490 kb eRNA. Out of those three, two shRNA showed significant decrease in MYC eRNA and concomitant decrease in MYC mRNA level. Now, we have added RT-PCR data for total four shRNA molecule against MYC-490 kb eRNA to strengthen our claim (Fig 2C) and to prove there was no off-target effect leading to the phenotype that we have described in our manuscript.

2. The authors should evaluate whether the MYC eRNA is enriched in a YEATS2 IP relative to control IgG. Controls of other eRNAs should be included that address sequence specificity.

We have conducted UV RNA immunoprecipitation (UV-RIP) using YEATS2 and control IgG antibodies, followed by qPCR analysis of MYC-490kb eRNA. Our results show that MYC eRNA is significantly enriched in the YEATS2 IP compared to the IgG control, indicating a specific association between MYC- 490kb eRNA and YEATS2. Modified figures are now included in Figure 3A of the revised manuscript.

To address sequence specificity, we also analysed other eRNA, MYC-425kb eRNA. We did not find any significant enrichment in the YEATS2 IP with TNF α stimulation, confirming the specificity of the MYC-490 eRNA-YEATS2 interaction. These results are now presented in Supplementary Figure 3D.

3. Bold claims are made by the authors about tyrosine phosphorylation regulating the eRNA:YEATS2 complex, yet experimental evidence directly shows this. The authors should compare wild-type YEATS2 to Tyr mutant in pulldown experiments with the eRNAs.

We thank the reviewer for the concern. As the transfection efficiency in the pancreatic cancer cells are very low, we have performed the lenti-virus mediated transduction of Wild type or mutant YEATS domain protein into MIAPaCa-2 cell to introduce YEATS domain containing construct and the mutant. Unfortunately, we could not detect any protein over-expression in the MIAPaCa-2 cells.

However, we have shown higher association of eRNA with YEATS2 protein in MIAPaCa-2 as well as in AsPc-1 cells (two different pancreatic cancer cell-lines) with TNF stimulation. We have shown dephosphorylation of YEATS2 in both MIAPaCa-2 and AsPc-1 cells with TNF induction. In the normal pancreatic epithelial cells- HPNE, where TNF-induced dephosphorylation event was not happening, we could not detect any increase in association of MYC eRNA with YEATS2 protein

there. These cellular data strongly proved our claim that the Tyr dephosphorylation event of YEATS2 is regulating the MYC eRNA binding in TNF-stimulated pancreatic cancer cells.

Moreover, by *in vitro* Phosphatase assay and by *in vitro* RNA binding assay (Fig 3K-3L), we have proved that the phospho-null mutant of YEATS2 (Y313F mutant) neither showed any decrease in Tyr phosphorylation level nor showed any increase in MYC eRNA association with the incubation of TNF-induced MIAPaCa-2 cell lysate. Based on all these data, we believe that the dynamic dephosphorylation event in Tyr moiety of YEATS2 protein is regulating the MYC eRNA binding in cancer cells.

4. Perform ChIP-qPCR with YEATS2 +/- eRNA knockdown and comparing wild-type YEATS2 to the tyr mutant.

We have already performed the ChIP-qPCR from MYC-490 kb eRNA knocked down cells to support our claim that MYC eRNA augmented the recruitment of YEATS2-containing ATAC complex into the MYC promoter and enhancer region (Fig 4E).

However, since we were unable to overexpress the wild type or Tyr mutant YEATS2 protein in MIAPaCa-2 cells, we could not perform the ChIP experiments in those conditions. But our *in vitro* RNA binding assay showed no increase in RNA association for the Y313F mutant (Fig 3L), indirectly proving that this mutant would show less association with the chromatin.

5. In Fig 2D, it is strange that 100 ng MYC-490 eRNA has no effect on MYC expression, 200 ng massively upregulates MYC expression, and 300 ng does not significantly increase MYC expression. To improve the confidence in this assay, the authors should treat cells with intermediate concentrations of MYC-490 eRNA (150 ng, 175 ng, 225 ng, 250 ng) to evaluate if these concentrations similarly upregulate MYC expression in their system.

We appreciate the reviewer's detailed observation regarding the dose-response of MYC-490 eRNA on MYC expression. We have conducted additional experiments using intermediate concentrations (150 ng, 175 ng, 225 ng, and 250 ng) as suggested by the reviewer to evaluate the consistency of MYC mRNA upregulation. Our findings confirm that MYC mRNA is significantly upregulated at 200 ng, while intermediate concentrations show a gradual increase but not to the same extent as 200 ng. These results suggest that the number of MYC-490kb eRNA produced from 200ng of plasmid DNA, might be mimicking the physiological stoichiometric ratio to increase MYC transcription significantly. This data has been incorporated into the revised Figure 2D (Lower panel).

6. To support the main claim of this study, it is critical that the authors knockdown/knockout YEATS2 and evaluate whether this alters MYC transcription in their pancreatic cancer models.

We thank the reviewer for the valuable suggestion. In response, we performed knockdown of YEATS2 using four different shRNA molecules in MIAPaCa-2 cells. Our results showed a significant reduction in MYC mRNA levels following YEATS2 knockdown, indicating that YEATS2 is essential for MYC transcription in our pancreatic cancer model. These findings have been added to Figure 3E and have been discussed in the revised Results section.

Referee #2:

The manuscript by Roy et al reports that MYC transcription in pancreatic tumor cells is regulated in part by MYC enhancer RNAs (eRNAs), especially when treated with TNF- α . The authors also show MYC-490 kb eRNA is elevated in chronic pancreatitis patients. The main thrust of this study was to examine how an eRNA (MYC-490 kb eRNA) influences MYC transcription. Overall, this study provides some new and interesting insights into the roles of MYC eRNAs by linking them to YEATS2 and TNF- α elevated MYC transcription. The authors also report that tyrosine phosphorylation of YEATS2 is decreased by TNF- α , which enhances the ability of MYC-490 kb eRNA to bind to YEATS2. However, as discussed below, the story is incomplete, and several issues are not adequately addressed.

Major Concerns:

1. This study refers to MYC-490 kb eRNA in the Results section. However, in the Discussion, the authors refer to it as MYC-490 kb eRNAs in at least one instance. Also, the authors show that MYC-425 kb eRNA as well as MYC-490 kb eRNA is upregulated when MIAPaCa-2 cells were treated with TNF- α . MYC-490 kb eRNA and MYC-425 kb eRNA are each likely to be a group of eRNAs expressed from their respective enhancer regions. The authors need to clarify this issue for the reader. If the authors believe that MYC-490 kb eRNA is a single eRNA and MYC-425 kb eRNA is a single eRNA, what are the sequences? Also, super enhancers comprise multiple enhancers distributed in some cases over many kb. For example, the upstream MYC super enhancer in colorectal tumor cells has been reported to comprise at least five MYC enhancers distributed over 40 to 50 kb. Are there multiple MYC eRNAs expressed from multiple MYC enhancers that are functional in TNF- α treated MIAPaCa-2 cells? In this regard, the authors identified the MYC super enhancer from data acquired in SW480 colorectal cells. A detailed discussion of MYC super enhancers is needed to provide the reader with a better perspective on MYC super enhancers.

We thank the reviewer for the valuable comments. We appreciate the opportunity to clarify the details regarding MYC-490 kb and MYC-425 kb eRNAs and their relation to the MYC super-enhancer region.

In our study, we analyzed publicly available ChIP-seq data for MIAPaCa-2 cells but took a novel approach of nascent RNA sequencing from EU-labelled RNA of MIAPaCa-2 cell using Click-chemistry to identify eRNAs originated from the MYC super-enhancer region. This analysis revealed that MYC-490 kb eRNA and MYC-425 kb eRNA are part of a group of 19 eRNAs expressed from a 260 kb region (chr8: 127163754–127428551) of the MYC super-enhancer. This region contains clusters of 18 MYC enhancers, with lengths ranging from 1 kb to 20 kb. We selected MYC-490 kb and MYC-425 kb eRNAs as representative eRNAs from this super-enhancer cluster, as they were specifically upregulated under chronic inflammatory condition. The coordinates of MYC-490 kb eRNA and MYC-425 kb eRNA are chr8:127247694-127248337 and chr8:127310599-127311551, respectively. We have included this part in detail in the revised manuscript.

Regarding the reference to SW480, we used this cell line as a point of comparison because the data available from SW480 cells were generated under chronic inflammatory conditions, similar to the context of our study, though in a different cellular background. We hope this explanation clarifies the relationship between MYC-490 kb eRNA, MYC-425 kb eRNA, and the MYC super-enhancer region described in our study.

2. The authors report that transfection of cells with an shRNA to knockdown MYC-490 kb eRNA reduced both MYC-490 kb eRNA and MYC mRNA to the same extent (Fig 2C). This finding raises two fundamental questions. First, this should only occur if there is a single key MYC eRNA, MYC-490 kb eRNA, that controls MYC transcription. While it may be the case, this seems highly unlikely for reasons discussed above (multiple MYC enhancers and multiple eRNAs). Also, about a 40-fold increase in MYC-490 kb eRNA (after transfection) led to only a 60 or 70% increase in MYC mRNA (Fig 2D). How do the authors explain these differing results. Also, the authors need to show how the transfection of the scrambled shRNAs effected on the levels of MYC-490 kb eRNA and MYC mRNA. Second, the authors report that they transfected the cells with Lipofectamine 3000. What was the transfection efficiency of the MIAPaCa-2 cells with the plasmids used in this study? PDAC cells transfect poorly, usually below 20%. This is key to interpreting the results of their transfection experiments.

We appreciate the reviewer's thoughtful comments. Our findings showed that knockdown of MYC-490 kb eRNA reduced both MYC-490 kb eRNA and MYC mRNA to a similar extent, however we do not claim that MYC-490 kb eRNA is the sole enhancer RNA regulating MYC gene transcription. The MYC locus is known to be regulated by a network of multiple enhancers and eRNAs. While MYC-490 kb eRNA seems to play a significant role in our experimental conditions, it is likely that other enhancers and eRNAs also contribute to MYC gene regulation, which needs further investigation.

The observed ~40-fold increase in MYC-490 kb eRNA and only a 60-70% increase in MYC mRNA likely reflects the complex nature of enhancer function. eRNAs often act as regulators that enhance transcriptional activity rather than directly dictating transcriptional output. The limited increase in MYC mRNA may result from additional layers of transcriptional regulation, including promoter saturation, feedback mechanisms, or post-transcriptional regulation.

Indeed, the transfection efficiency of MIApaCa-2 and other pancreatic cancer cells were very poor. We suffered a lot in transfecting the cells and hence could not over-express the wild type or Y313F mutant of YEATS domain protein to perform two experiments as suggested by Reviewer 1. However, the shRNA transfection was considerably better, so we got significant knock-down of the target MYC eRNAs or YEATS2 mRNA. We have included data showing that transfection with scrambled shRNA did not significantly affect the levels of MYC-490 kb eRNA or MYC mRNA (Fig 2C or 3E).

3. The authors should check by CHIP-qPCR, the association of the ATAC complex with the MYC-425 enhancer region

We thank the reviewer for the suggestion. We performed CHIP-qPCR using a ZZZ3 antibody to assess the association of the ATAC complex with the MYC-425 enhancer region. Our results show an enrichment of the MYC-425 enhancer region with YEATS2-containing ATAC complex. This data (Supplementary Fig S4A) and related clarification has been included in the revised manuscript.

4. A significant limitation of this study is that it reports the results with only one PDAC cell line. Are their findings generally true for PDAC cells or specific to MIApaCa-2 cells? It would be helpful to know if any of the major findings occur in other PDAC cell lines. For example, does TNF- α increase the expression of MYC-490 kb eRNA and MYC-425 kb eRNA in other PDAC cell lines? Also does TNF- α increase the association of the ATAC complex with the MYC enhancer in other PDAC cell lines?

We thank the reviewer for raising this important point regarding the broader applicability of our findings across pancreatic ductal adenocarcinoma (PDAC) cell lines. To address this, we performed additional experiments using the AsPC-1 cell line, another well-established PDAC model. Consistent with our observations in MIAPaCa-2 cells, treatment with TNF- α led to a significant upregulation of both *MYC*-490 kb eRNA and *MYC*-425 kb eRNA expression in AsPC-1 cells (Figure 1D). We have observed higher association of *MYC*-490 kb eRNA with YEATS2 protein as well as decrease in Tyrosine phosphorylation level of YEATS2 protein in TNF-induced AsPC-1 cells. Furthermore, TNF- α treatment increased the association of the ATAC complex with the *MYC* enhancer region in AsPC-1 cells, demonstrating a similar pattern to that observed in MIAPaCa-2 cells earlier. We have incorporated all the data into the revised manuscript.

5. In the discussion, the authors argue that "it is possible that in chronic inflammatory condition, the *MYC* eRNA confers the specificity of YEAT2 towards crotonyl-lysine binding over acetyl-lysine binding." However, the authors probed for H3K27cr and found no evidence for it. Thus, the authors should omit this commentary until they have data to support it.

We thank the reviewer for insightful comment. We acknowledge the concern regarding the lack of direct evidence for H3K27cr in our study. In response, we have revised the discussion to remove speculative statements about the role of *MYC* eRNA in conferring YEATS2 specificity towards crotonyl-lysine binding. Instead, we have refined our interpretation to align more closely with the available data.

6. The effect of *MYC*-490 kb eRNA on proliferation is very small or non-existent. The difference between the vector control and the *MYC*-490 kb eRNA transfected cells does not appear to be statistically significant.

We appreciate the reviewer's observation regarding the modest impact of *MYC*-490 kb eRNA on cell proliferation. We agree that the overexpression of a single enhancer RNA may not be sufficient to induce a significant proliferative response. Enhancer RNAs often act as part of a broader regulatory network, influencing gene expression through multiple interactions rather than acting independently. This may explain why we did not observe a significant difference between the vector control and *MYC*-490 eRNA transfection groups. However, we have changed the representation to make it more convincing to the reader.

Minor Concerns:

1. A better/additional reference regarding *MYC* enhancers activated in tumor cells is Dave et al eLife 6, e23382 (2017).

Thank you for the suggestion. We have incorporated the reference (Dave et al., eLife 6, e23382, 2017) into the main manuscript as recommended.

2. The authors should provide the reference for the GRO-Seq data mentioned in their study - reference 32?

We thank the reviewer for pointing this out. We have removed the GRO-Seq analysis from our study, and reference 32 has been updated accordingly. Instead, we have used our own lab generated nascent RNA-sequencing data (Fig 1B) for TNF-treated or untreated MIAPaCa-2 cells.

3. The data in Fig S1E is not mentioned in the manuscript. Either discuss the data or omit it from the manuscript.

We thank the reviewer for their comment. However, there was no Fig. S1E in the submitted manuscript; the supplementary figures only go up to S1D. However, in the revised manuscript, we have added new data as Fig. S1E and it is now appropriately discussed in the text.

4. The decreased expression of MYC-490 kb eRNA reported in Fig 2C left panel and MYC mRNA in Fig 2C right panel is not nearly 50 or 40 to 50%, respectively.

We thank the reviewer for this insightful observation. Upon re-assessing the data, we acknowledge that the reduction in MYC-490 kb eRNA expression (Fig. 2C, left panel) and MYC mRNA expression (Fig. 2C, right panel) is approximately 30%, rather than the previously stated 50% or 40–50%. We have revised the corresponding text in the manuscript to ensure the data are accurately described.

5. Why did the authors select YEATS2 for their CHIP studies? Many other proteins, including p300 are predicted to bind MYC-490 kb eRNA, and p300 has been shown to associate with the MYC super enhancer.

We thank the reviewer for the concern. Having the list of MYC-490 kb eRNA associated proteins from our *in-silico* study, we performed UV-RIP experiments with BRD4 and p300 proteins to check their association with MYC eRNAs as they are well known eRNA binding proteins. However, we could not detect any significant association of MYC eRNAs with those proteins in TNF-stimulated MIAPaCa-2 cells. In contrast, we detected significant higher association of YEATS2 protein with MYC eRNAs in TNF-stimulated MIAPaCa-2 cells. So, we selected YEATS2 protein for our subsequent studies.

6. The authors refer to H3 in Fig 4E. Was the H3K27ac antibody used? If so, this should be spelled out.

We thank the reviewer for their comment. In Fig. 4E, only the histone H3 antibody was used, not the H3K27ac antibody. We have clarified this in the revised manuscript to avoid any confusion.

7. The font of many of the figures is too small to read.

In the revised manuscript, we have increased the font size in all figures to ensure better readability.

8. CBP/p300 function primarily as histone writers not as histone readers.

We thank the reviewer for the comment. We have revised the manuscript to clarify that CBP/p300 primarily function as histone writers rather than histone readers.

9. The authors claim that some of their data "proves" their hypothesis. The data that they are referring to is consistent with their hypothesis. You can test a hypothesis, but not prove a hypothesis.

We thank the reviewer for the helpful comment. We have updated the manuscript to replace the word "proves" with "supports" to more accurately describe how our data align with the hypothesis, rather than proving it definitively.

Dear Dr. Mazumder,

Thank you for the re-submission of your revised manuscript. We have now received the enclosed report from referee 1, and I am happy to say that this referee supports the publication of your ms now. Referee 2 was unfortunately not available to re-review your ms, but referee 1 has also assessed your response to referee 2's comments.

Only a few editorial requests will need to be addressed before we can proceed with the official acceptance of your study:

- Your ms has 5 main figures and should thus be published as a short report with combined results and discussion sections and a maximum of 29,000 characters (excluding methods and references). Please refer to our guide to authors online for more specific information.

- Please upload the ms file as a word file without figures.

- Please add a Data Availability Section (DAS) to the end of the methods. The DAS needs to provide the specific URL for the GSE288088 dataset and this dataset needs to be freely accessible upon online publication of your ms.

- The conflict of interest statement is missing the section heading "Disclosure and Competing Interests Statement"

- The author credits need to be removed from the ms file. All credits are entered during online ms submission.

- The REFERENCE format needs to be alphabetical, not numerical; et al needs to be used after 10 author names. Please correct to the EMBO reports reference style.

- Please send us with your final ms a completed author checklist, which you can download from our author guidelines <<https://www.embopress.org/page/journal/14693178/authorguide>>. The completed author checklist will also be part of the transparent peer-review file (RPF).

- The FUNDING INFO is not congruent: 201610079104 and RCB/NIBMG-Ph.D./2021-22/M/278/1003 are missing in our online submission system as grant numbers; all grant numbers there are "000" - if there aren't any grant numbers it is fine to leave the blank space(s), otherwise, all funders and their grants acknowledged in the ms need to be entered in the system so that the info matches.

- The main figures need to be uploaded as separate production quality Figure files, their legends should be provided at the end of the ms.

- The supplementary figures and tables should be called Expanded View. You can upload them as Figure EV1-4 and as Table EV1-X. Some of the tables can be moved to the Reagents and Tools table (see below). The EV figures and tables need to be uploaded as individual files. The EV legends need to be in the ms file after the main figure legends, and the table legends need to be within the table files. All ms callouts need to be updated accordingly.

- The Methods section should include a Reagents and Tools Table (listing key reagents, experimental models, software and relevant equipment and including their sources and relevant identifiers) and a Methods and Protocols section in which the methods should be described using a step-by-step protocol format with bullet points, to facilitate the adoption of the methodologies across labs. More information on how to adhere to this format as well as downloadable templates (.docx) for the Reagents and Tools Table can be found in our author guidelines: <<https://www.embopress.org/page/journal/14693178/authorguide#manuscriptpreparation>>.

- The Abbreviations section needs to be removed from the manuscript. Abbreviations should be defined in brackets after their first mention in the text, not in a list of abbreviations.

- The Significance section on the title page needs to be removed.

- Materials and Methods should be just Methods.

Figure Legends - Comments

- Please define the annotated p values ****/***/**/* as well as provide the exact p-values for the same in the legend of supplementary figure 4A as appropriate.

- Please note that the exact p values are not provided in the legends of figures 2A, C, D; 3A, B, E, J, L; 4B, C, E; 5B; Supplementary figure(s) 1B-D; 2A; 3C, D. Please provide the exact p-values as reasonable.

- Please note that in figures 4B, C, E; 5B; Supplementary figure(s) 1B-D there is a mismatch between the annotated p values in

the figure legend and the annotated p values in the figure file that should be corrected.

- Please note that information related to n is missing in the legend of figure 5A

I would like to suggest a few minor changes to the abstract that needs to be written in present tense. Please let me know whether all is correct and whether you agree with the following:

Pancreatic ductal adenocarcinoma (PDAC) is one of the most prevalent and aggressive forms of pancreatic cancer with low survival rates and limited treatment options. Aberrant expression of the MYC oncogene promotes PDAC progression. Recent reports have established a role for enhancer RNAs (eRNAs), originating from active enhancers, in controlling gene transcription. Here we show that a novel MYC eRNA regulates MYC gene expression during chronic inflammatory conditions in pancreatic cancer cells. A higher amount of MYC eRNA is observed in chronic pancreatitis as well as in pancreatic cancer patients. We show that MYC eRNA interacts with YEATS2, a histone reader protein of the ATAC-HAT complex, and augments the association of YEATS2-containing ATAC complexes with MYC promoter/enhancer regions and thus increases MYC gene expression. TNF- α induced Tyrosine dephosphorylation of the [OK?] YEATS domain regulates [promotes or decreases??] MYC eRNA binding to the YEATS2 protein in pancreatic cancer cells. Our study adds another regulatory layer of MYC gene expression by enhancer-driven transcription.

EMBO press papers are accompanied online by A) a short (1-2 sentences) summary of the findings and their significance, B) 2-3 bullet points highlighting key results and C) a synopsis image that is exactly 550 pixels wide and 200-600 pixels high (the height is variable). The synopsis image should provide a sketch of the major findings, like a graphical abstract. Please note that text needs to be readable at the final size. Please send us this information along with the final manuscript.

Referee #1:

After reviewing the revised manuscript, I have come to the conclusion that the authors have made sufficient efforts to improve the technical deficiencies of the work. I support publishing this.

I also looked at how the authors responded to referee 2's comments and I think they did enough to address them.

All editorial and formatting issues were resolved by the authors.

Dr. Anup Mazumder
BRIC-National Institute of Biomedical Genomics
P.O.: N.S.S
Kalyani, West Bengal 741251
India

Dear Dr. Mazumder,

I am very pleased to accept your manuscript for publication in the next available issue of EMBO reports. Thank you for your contribution to our journal.

Yours sincerely,
